# Climate evolution across the Mid-Brunhes Transition

Aaron M. Barth[1,2*], Peter U. Clark[2], Nicholas S. Bill[2], Feng He[3,4], Nicklas G. Pisias[2]

[1]Department of Geography and Earth and Environmental Sciences, Emory and Henry College, Emory, VA 24327, USA

[2]College of Earth, Ocean, and Atmospheric Sciences, Oregon State University, Corvallis, OR 97331, USA

[3]Department of Geoscience, University of Wisconsin – Madison, Madison, WI 53706, USA

[4]Center for Climatic Research, Nelson Institute for Environmental Studies, University of Wisconsin – Madison, Madison, WI 53706, USA

*Correspondence to:* Aaron M. Barth (abarth@ehc.edu)

**Abstract.** The Mid-Brunhes Transition (MBT) began ~430 ka with an increase in the amplitude of the 100-kyr climate cycles of the past 800,000 years. The MBT has been identified in ice-core records, which indicate interglaciations became warmer with higher atmospheric $CO_2$ levels after the MBT, and benthic oxygen isotope ($\delta^{18}O$) records, which suggest that post-MBT interglaciations had higher sea levels and warmer temperatures than pre-MBT interglaciations. It remains unclear, however, whether the MBT was a globally synchronous phenomenon that included other components of the climate system. Here we further characterize changes in the climate system across the MBT through statistical analyses of ice-core and $\delta^{18}O$ records as well as sea-surface temperature, benthic carbon isotope, and dust accumulation records. Our results demonstrate that the MBT was a global event with a significant increase in climate variance in most components of the climate system assessed here. However, our results indicate that the onset of high-amplitude variability in temperature, atmospheric $CO_2$, and sea level at ~430 ka was preceded by changes in the carbon cycle, ice sheets, and monsoon strength during MIS 14 and 13.

## 1 Introduction

The last 800 kyr of the Pleistocene epoch is characterized by the emergence of dominant ~100-kyr glacial-interglacial climate cycles (Pisias and Moore, 1981; Imbrie et al., 1993; Raymo et al., 1997; Clark et al., 2006). These climate cycles typically have long glacial periods punctuated by short interglaciations. Since ~430 ka (i.e., starting with Marine Isotope Stage (MIS) 11), interglaciations have experienced warmer temperatures (Jouzel et al., 2007) and higher concentrations of atmospheric $CO_2$ (Luthi et al., 2008) relative to earlier interglaciations of the last 800 kyr (Figure 1). The transition to higher amplitude interglaciations has also been recognized in deep-sea records of $\delta^{18}O$ measured in benthic foraminifera (Lisiecki and Raymo, 2005) that identify lesser ice volume and/or warmer deep-ocean temperatures (Figure 1).

Jansen et al. (1986) originally described this change in amplitude of interglaciations as a singular Mid-Brunhes Event, but Yin (2013) argued that it is more appropriately considered as a transition between two distinct climate states, thus referring to it as the Mid-Brunhes Transition (MBT). The change from low-amplitude to high-amplitude 100-kyr variability at ~430 ka occurs during an interval of reduced eccentricity and corresponding precession (Figure 1), but similar orbital forcing occurred at times before and after the onset of the MBT with no comparable response, suggesting that the MBT was an unforced change internal to the climate system. Mechanisms proposed for the MBT include a latitudinal shift in the position of the Southern Hemisphere westerlies that increased upwelling of respired carbon in the post-MBT Southern Ocean (Kemp et al., 2010), and a change in Antarctic Bottom Water (AABW) formation through insolation-induced feedbacks on sea ice and surface water density (Yin, 2013). However, several questions remain. (1) How and when was the MBT expressed in other components of the climate system? (2) Was the MBT a global or regional transition? (3) Did components expressing a transition change synchronously? Here we address these questions by providing a statistical characterization of changes occurring over the last 800 kyr as recorded by a variety of paleoclimatic proxies with broad spatial coverage.

## 2 Methods

### 2.1 Data collection

We compiled all available published records of sea-surface temperature (SST), benthic marine carbon isotopes ratios ($\delta^{13}$C), and dust accumulation (Dust) that met our selection criteria and closely represented a global distribution as attainable (Figure 2). Each data set has an average temporal resolution of <5 kyr, does not include any large age gaps, and spans much or all of the entire time period of consideration to limit biasing of the younger parts of the record. Lisiecki (2014) placed all of the $\delta^{13}$C records on the LR04 age model. Published SST records that were not on the LR04 age model were placed on it in one of two ways. If the original data had depth and benthic $\delta^{18}$O data, the SST record was placed on LR04 using the ager script in MATLAB as part of the ARAND software package (Howell et al., 2006). When only benthic $\delta^{18}$O records were available, the SST records were placed on LR04 by selecting corresponding tie points in the $\delta^{18}$O data series using the AnalySeries version 2.0 software (Paillard et al., 1996). Because some dust records could not be placed on the LR04 age model, certain statistical analyses of them (e.g., phase/lag relationships) are likely not robust, but the overall variance in them is preserved. Each record was then interpolated to a time step ($\Delta$t) of 2 kyr. With each record having an average resolution <5 kyr, this $\Delta$t allows for the preservation of higher frequency variability while limiting the number of interpolated data points.

We used empirical orthogonal function analysis (EOF) to characterize the dominant modes of variability and robustly demonstrate global and regional signals of the SST, $\delta^{13}C$, and dust records. We then used spectral analyses of each resulting principal component (PC) to characterize their periodicity, phase, and amplitude.

## 2.2 Sea-surface temperatures

We used 11 SST records that span the entire 800-kyr time period, and four additional records that span 8 – 758 ka. Inclusion of these four shorter records does not change our conclusions. The SST records cover the Pacific (n = 9), Atlantic (n = 5), and Indian (n = 1) Oceans (Figure 2, Table 1). We note that Shakun et al. (2015) reconstructed a global SST stack for the last 800 kyr using 49 records, but only seven of these spanned the entire 800 kyr. Comparison of our SST PC1 based on 15 records to the Shakun SST stack shows excellent agreement (Fig. S1).

## 2.3 Carbon isotopes ($\delta^{13}C$)

We analyzed the global set of $\delta^{13}C$ records compiled by Lisiecki (2014) (n = 26; Figure 2), and separately analyzed the records in the Atlantic (n = 14) and the Pacific (n = 4) basins, thus distinguishing between the dominant water masses within each basin and removing the muting effect of the more negative Pacific values on the more positive Atlantic. Similar to SSTs, Lisiecki (2014) reconstructed a global $\delta^{13}C$ stack for the last 3 Myr using 46 records, but only 18 of these spanned the last 800 kyr. Comparison of our $\delta^{13}C$ PC1 to the Lisiecki $\delta^{13}C$ stack shows excellent agreement (Fig. S2).

We then looked at regional and depth stacks of the $\delta^{13}C$ records in the Atlantic basin to characterize changes in the dominant water masses on orbital time scales. Regional stacks were broken into North Atlantic (> 20º N; n = 4), equatorial Atlantic (20º S to 20º N; n = 14), and South Atlantic (> 20º S; n = 8). We also created stacks for the deep North Atlantic (depth > 2000 m; n = 4) and intermediate North Atlantic (depth < 2000 m; n = 3). All included records were averaged to create the stack and each stacked record was interpolated to a 2-kyr-time step. Stacking improves the signal-to-noise ratio of the $\delta^{13}C$ records, making regional stacks useful in identifying circulation changes and comparing circulation responses with other climate records (Lisiecki, 2014).

## 2.4 Dust

We analyzed eight proxy records of dust that span the entire 800-kyr time period, and then separated them by hemisphere (Northern = 3, Southern = 5) to characterize hemispheric differences (Figure 2). The various proxies for dust include Fe mass accumulation rates, weight percent of terrigenous material and Fe, flux of lithogenic grains, and grain size analysis. We standardized each record

before analysis to account for these various proxy types and their differing range in values, thus allowing for comparison of their relative amplitudes of variation.

**2.5 Empirical Orthogonal Function analysis (EOF)**

We used EOF analysis to objectively characterize the climate variability recorded by the proxies across the MBT. Analyses of covariance between the data were conducted using the EOF script as part of the ARAND software package (Howell et al., 2006). The results provide both the dominant variability as a time series (principal component) and a spatial distribution of variance contribution (factor loadings). The records for SST and $\delta^{13}C$ were kept in their original values of degrees and per mil, respectively, to preserve the original variance. Dust records were standardized to a mean value of zero and unit variance so that each record provided equal weight to the EOF. Statistical significance of all EOFs was determined through segmented linear regression analysis. All resulting break points occur on or after the second EOF and are thus considered significant.

**2.6 Spectral analysis**

We used the Blackman-Tukey technique in the ARAND software package for spectral analysis of each PC (Howell et al., 2006). Analyses were conducted using all data points within the time interval of interest, boxcar windowing of the input data, and hamming spectral filter. Multiple tests were conducted for the time slices 8-800 ka, 450-800 ka, and 8-350 ka. These intervals characterize the dominant frequency of variability over the entire 800-kyr record, and for the pre- and post-MBT intervals, respectively. The removal of the 350-450 ka interval limited the influence of MIS 11, MIS 12, and Termination V (T5) as these were shown to potentially bias the spectral power. Furthermore, these selected intervals result in time series of equal length to limit biasing of longer records. Additional tests were conducted using wavelet analyses that characterize the change in spectral power as a time series. Complementary spectral analyses were conducted on $CO_2$ and $CH_4$ records from the EPICA Dome C ice core (EPICA-community-members, 2004; Jouzel et al., 2007), and benthic $\delta^{18}O$ using the LR04 stack (Lisiecki and Raymo, 2005). Cross-spectral analyses were conducted for the PCs against mean insolation values to determine phase and coherency of each. Mean insolation values were calculated for each of the dominant periodicities (eccentricity, obliquity, and precession) with the data derived from AnalySeries (Laskar et al., 2004; Paillard et al., 1996).

**2.7 Variance tests**

We used f-tests to test for variance changes across the MBT for each principal component from the EOF analysis as well as for $CO_2$, $CH_4$, and the LR04 $\delta^{18}O$ records. Analyses were conducted in MATLAB using the *vartest2* script. This approach assumes

the null hypothesis that the pre- and post-MBT distributions of the time series of each climate component have the same normally distributed variance. If the resulting variance values reject this hypothesis of no statistical difference, then the pre- and post-MBT time series are determined to have undergone a significant change in variance across the MBT. We interpret the change in variance to reflect a change in the amplitude of each climate signal.

## 3 Results

### 3.1 $CO_2$, $CH_4$, and benthic $\delta^{18}O$

Time series of the greenhouse gases $CO_2$ and $CH_4$ and of the LR04 stack of benthic $\delta^{18}O$ suggest an increase in their interglacial values across the MBT (Figure 1). Spectral analyses of the LR04 stack and atmospheric $CO_2$ indicate a small post-MBT increase

in the 100-kyr band, whereas results for $CH_4$ indicate a decrease (Figure S3). All three records show an increase in the precessional band (19-23 kyr). Variance tests suggest that $\delta^{18}O$ and $CO_2$ have a statistically significant increase in variance across the MBT while $CH_4$ variance decreases (Table S1).

### 3.2 Sea-surface temperatures

EOF analysis of global SSTs over the last 758 kyr identifies two statistically significant principal components (Figure 3a). The first and second principal components (PC1 and PC2, respectively) account for 69% of the total variance with PC1 explaining 49% alone. While some degree of regional variability in each record exists, factor loadings indicate that each record positively contributed to PC1 with a larger contribution coming from high-latitude records. Thus, PC1 is representative of a global SST signal. SST PC1 demonstrates a stepwise increase in variance starting at 436 ka, with an increase of interglacial temperatures while

showing no significant change in the lower limit glacial values, which is one of the defining characteristics of the MBT. The highest spectral density is in the 100-kyr-frequency band throughout the entire time period (Figure S3d). Wavelet analysis (Figure 4a) shows a significant increase in the 100-kyr-frequency band 580 ka that reaches its maximum spectral power during MIS 11 and persists throughout most of the remaining interval, albeit with decreasing intensity after ~250 ka. Variance f-tests reveal a significant increase in amplitude from the pre- to post-MBT SSTs (Table S1). These results thus confirm that there was a stepwise

global transition of SSTs from lower to higher amplitude interglaciations as previously inferred from individual records.

Variance calculations on proxies of bottom water temperature (Elderfield et al., 2012) and on the Antarctic EPICA ice-core deuterium record (EPICA-community-members, 2004), a measure of Antarctic atmospheric temperature, also indicate statistically

significant increases in variance across the MBT (Table S1). In both proxies, the time series indicate an increase of interglacial temperature values while showing no significant change to the lower limit glacial values, similar to PC1 of SSTs (Figure 5).

### 3.3 Dust

The EOF analysis of the global dust records identifies two statistically significant principal components with PC1 representing 56% of the total variance and PC2 15% (Figure 3b). All records but the one from the Chinese Loess Plateau (CLP) reflect increased dust accumulation due to increased aridity and/or wind strength during glaciations, whereas higher dust accumulation in the CLP record reflects increased summer Asian monsoon strength, which is an interglacial signal (Sun and An, 2005). Accordingly, factor loadings for the dust records are all positive for PC1 except for the CLP.

In contrast to the change in variance seen in temperature, $CO_2$, and $CH_4$ during MIS11, variance tests of the dust PC1 suggest a stepwise increase in variance during MIS12, with subsequent glaciations having higher amplitudes (Table S1). Separating the records by hemisphere shows that the increase in glacial amplitude starting at MIS 12 occurs in the southern PC1 but not in the northern PC1 (Figure 6). Similarly, the signal during MIS 14 present in the global PC1 is absent in the northern PC1, suggesting that the northern control on dust accumulation was skipped during that glacial.

Spectral analysis of the global PC1 indicates dominant power in the 100-kyr-frequency band that increases in spectral power across the MBT (Figure S3b). Furthermore, wavelet analysis of PC1 demonstrates an increase in the spectral power of the 100-kyr band at ~600 ka with its highest power during MIS 11 (Figure 4b), similar to the SST PC1. The 100-kyr frequency remains statistically significant throughout the interval 100-600 ka.

### 3.4 $\delta^{13}$C

The first principal component of the global $\delta^{13}$C ($\delta^{13}C_G$; PC1) explains 58% of the total variance (Figure 3c). EOF analysis of $\delta^{13}$C records from the Atlantic basin ($\delta^{13}C_{ATL}$) yields two statistically significant PCs with PC1 and PC2 explaining 58% and 13% of the total variance, respectively (Figure 3d). EOF analysis of $\delta^{13}$C records from the Pacific ($\delta^{13}C_{PAC}$) yields one statistically significant principal component (PC1 = 81% total variance) (Figure 4e).

Both the global and Atlantic PC1 exhibit a strong 100-kyr frequency that is persistent from 680 ka to 180 ka (Figure 4c, 4d). Unlike SST and dust, however, $\delta^{13}C_G$ and $\delta^{13}C_{ATL}$ demonstrate a stronger 100-kyr power prior to MIS 11 with its highest power throughout

MIS 13 and 12 (510-460 ka). Spectral analysis shows a decrease in power of the 100-kyr-frequency band from pre- to post-MBT (Figure S3f, S3g). Variance tests show that the pre- and post-MBT intervals for $\delta^{13}C_G$ and $\delta^{13}C_{ATL}$ are statistically different with

175 higher variance during the pre-MBT (Table S1). Spectral analyses and variance tests of $\delta^{13}C_{PAC}$ PC1 are similar to $\delta^{13}C_G$ and $\delta^{13}C_{ATL}$ PC1s. The only difference between the three PC1s is there is less variance recorded in $\delta^{13}C_{PAC}$ (Figure 3e). We interpret this muted signal to be a result of three factors: the large size of the Pacific relative to the Atlantic, less mixing between water mass end members such as the positive NADW and more negative AABW, and ocean circulation aging the carbon isotopes over time leading to more homogenized water masses in the Pacific.

Factor loadings for $\delta^{13}C_{ATL}$ PC1 are all positive suggesting that the time series is representative of the entire Atlantic basin. In contrast, $\delta^{13}C_{ATL}$ PC2 yields negative values for all but the intermediate North Atlantic records and does not show strong 100-kyr spectral power. As such, these results suggest that PC2 exhibits the dominant mode of variability recorded in the benthic $\delta^{13}C$ of North Atlantic waters shallower than 2000 m depth. Curry and Oppo (2005) show that NADW formation to below ~2000 m is

185 reduced in the North Atlantic during glacial times. The sites with positive factor loadings in PC2 are located at depths < 2000 m, and therefore each site should remain consistently bathed in NADW through glacial-interglacial cycles. We thus interpret PC2 as a record of changes in the isotopic values of the North Atlantic carbon reservoir rather than circulation changes.

During MIS 13, all three $\delta^{13}C$ PC1s (global, Atlantic, and Pacific) demonstrate high positive values. This excursion, first recognized

in individual records by Raymo et al. (1997), clearly stands out relative to other $\delta^{13}C$ interglacial values recorded throughout the last 800 kyr. The MIS 13 excursion is even more apparent when compared against other proxy records such as atmospheric $CO_2$, SST, and $CH_4$ (Figure 7). This high-amplitude change in $\delta^{13}C$ values is similar to the changes recorded in other proxies during MIS 11, yet precedes the MBT by one glacial cycle. Removal of the MIS 13 interval from variance tests results in no statistical difference in variance before and after the MBT suggesting a large effect of the carbon isotope excursion on these calculations.

### 3.4 $\delta^{13}C$ gradients

Figure 8 shows regional stacks of $\delta^{13}C$ from the deep (>2000 m) and intermediate (<2000 m) North Atlantic and the deep South Atlantic. As discussed, the intermediate North Atlantic (INA) signal is predominantly controlled by changes in the carbon reservoir over orbital time scales. In contrast, the deep North Atlantic (DNA) is controlled by changes in the relative influence of isotopically

more positive NADW and isotopically more negative AABW, as well as any $\delta^{13}C$ changes to reservoir that feeds the deep basin from shallower and surficial waters (i.e., INA). Subtracting the INA from the DNA record (i.e. depth gradient) removes the

influence of reservoir changes, with the residual time series reflecting only the relative influences of AABW and NADW on the isotopic values of carbon in the deep North Atlantic. This is supported by comparing the North Atlantic depth gradient time series against the South Atlantic stack (Figure S4). Both time series demonstrate good correlation for the entire time interval ($r^2 = 0.58$), but even more striking is the similarity in $\delta^{13}C$ values, with both time series showing similar variability and range in $\delta^{13}C$ space. The isotopic similarity between the two records suggest adequate removal of reservoir influences with the North Atlantic depth gradient thus reflecting changes in dominant water mass influence (i.e. circulation). We also note that the correlation between the two records increases starting at MIS 15 (~530 ka).

The depth gradient does not show the prominent MIS 13 excursion that was present in the original DNA stack (Figure 8), suggesting that the excursion is likely due to a change in the carbon reservoir (represented by the INA) and not related to ocean circulation. Figure 9 shows contour $\delta^{13}C$ plots of the Atlantic basin for MIS 13 and MIS 5e. Although there is some uncertainty in the these plots due to limited spatial coverage, they show a clear enrichment of the entire basin during MIS 13 relative to average post-MBT interglacial conditions, as represented here by MIS 5e. The global and Pacific $\delta^{13}C$ PC1s also show the MIS 13 $\delta^{13}C$ excursion, suggesting a change in the global carbon reservoir.

We next evaluate the latitudinal gradient between the South Atlantic signal and the DNA signal in order to further assess the relative influence of the more negative AABW $\delta^{13}C$ values on North Atlantic $\delta^{13}C$ values (Figure 10). Lisiecki (2014) interpreted weaker gradients during glaciations to reflect shoaling of NADW and greater penetration of AABW, which could result from reduced NADW formation or stronger AABW formation. Figure 10b shows a stepwise drop in mean values beginning in MIS 12 (~436 ka), suggesting a weakening of the gradient due to greater similarity between North Atlantic and South Atlantic glacial and interglacial $\delta^{13}C$ values.

**4. Discussion**

Our new analyses demonstrate that there was a statistically significant increase in variance in atmospheric $CO_2$, Antarctic temperature, global SSTs, and bottom-water temperature at 436 ka. These changes are consistent with a transition between two distinct climate states associated with higher amplitude interglaciations starting with MIS 11, supporting the notion of a MBT as defined by Yin (2013). The same climate variables mentioned above also show an increase in spectral power in the 100-kyr-frequency band after the MBT. On the other hand, the dust analyses suggest that the transition to greater variability was experienced in the Southern Hemisphere in the glacial periods starting with MIS 12.

## 4.1 MIS 13 carbon isotope excursion

The PC1 of $\delta^{13}C_G$ shows a strong correlation with the $CO_2$ record for most of the last 800 kyr (Figure 7a). The exception is during MIS 13, when $CO_2$ levels were still at pre-MBT levels while $\delta^{13}C_G$ shows an anomalously high enrichment relative to other interglacial values. This is further illustrated by $\delta^{13}C$ contour plots showing that the Atlantic basin was enriched in $\delta^{13}C$ during MIS 13 relative to the MIS 5e (Figure 9).

We evaluated records of biologic activity in various locations of the Atlantic and Pacific Oceans to assess potential sources and sinks in the carbon system during MIS 13. Ba/Fe from the Antarctic Zone (AZ) records the sedimentary concentration of biogenic Ba and is thus a proxy of organic matter flux to the deep ocean south of the Polar Front (Jaccard et al., 2013), whereas alkenone concentrations from the Subantarctic Zone (SAZ) indicate export productivity to the deep ocean in the region north of the Polar Front (Martínez-Garcia et al., 2009). Based on these proxies, Jaccard et al. (2013) argued that there were two modes of export productivity in the Southern Ocean (SO), where high/low export occurs in the AZ during interglaciations/glaciations, and low/high export occurs in the SAZ during interglaciations/glaciations. They attributed the increase in SAZ export productivity to iron fertilization from increased dust accumulation in the SAZ associated with intensified SO westerlies during glacial periods. Our Southern Hemisphere dust PC1 record supports this hypothesis in showing that high values of dust accumulation correlate with increased values of SAZ export productivity over the last 800 kyr (Figure S5). We note, however, that the increase in dust starting at MIS 12 does not have an associated decrease in glacial $CO_2$ values, suggesting that if iron fertilization contributed to lower $CO_2$ levels, it had an upper limit beyond which additional dust fluxes had little effect.

The antiphase relationship between export productivity between the SAZ and AZ requires a mechanism to increase organic matter productivity in the AZ during interglaciations as suggested by the Ba/Fe signal (Figure S5c). In the modern SO, vertical mixing and upwelling drive the delivery of nutrient-rich waters necessary for biologic activity to the surface ocean. Wind-driven upwelling is associated with SO westerlies which shift poleward during interglaciations (Toggweiler et al., 2006). Thus, any reduction of upwelling would result from a more northerly position or decrease in strength of the westerlies; a further decrease in nutrient-rich surface waters in the AZ during glaciations likely resulted from increased SO stratification (Sigman et al., 2010; Jaccard et al., 2013). We note, however, that Jaccard et al. (2013) find no AZ export productivity during MIS 13 whereas all other interglaciations over the last 800 kyr show some evidence for it (Figure S5c). This skipped interglaciation in export productivity suggests some

combination of a change in the position/strength of the SO westerlies or stratification of the AZ that limited the delivery of nutrient-rich deep waters to the surface as compared to other interglaciations of the last 800 kyr.

The PC1s of $\delta^{13}C$ (global, Atlantic, and Pacific) demonstrate that the global ocean was enriched in heavy carbon during MIS 13 relative to any other interglaciation of the last 800 kyr (Figure 3). In contrast, atmospheric $CO_2$ concentrations were ~240 ppm during MIS 13, similar to other pre-MBT interglacial levels (Figure 1). Ba/Fe records of organic export productivity from the AZ that acts as a sink for light carbon indicate no increase during this interglaciation while Ca/Al records from the SAZ indicate increased preservation and thus a deeper lysocline and lower dissolved inorganic carbon (Jaccard et al., 2010). The question thus becomes: if the ocean is heavily enriched in $\delta^{13}C$ during MIS 13 while $CO_2$ and export productivity remained at low levels, what reservoir contained the isotopically light carbon?

Paleoclimate records from the CLP indicate greater precipitation during MIS 13 relative to the other interglaciations (Liu, 1985; Yin and Guo, 2008). This greater precipitation has been attributed to increased monsoon activity recognized throughout monsoonal areas of the Northern Hemisphere and persisting through MIS 15, 14, and 13 (Yin and Guo, 2008; Guo et al., 2009). Biogenic silica measurements from Lake Baikal exhibit continuously high terrestrial productivity in central Asia throughout MIS 15 to MIS 13 (Prokopenko et al., 2002), whereas sea-level reconstructions indicate that ice volume during MIS 14 was considerably less relative to other glacial maxima of the last 800 kyr (Figure 11d) (Elderfield et al., 2012; Shakun et al., 2015). Thus, the smaller ice sheets of MIS 14 would likely have had a lesser effect on displacing forested areas of the Northern Hemisphere, allowing greater terrestrial carbon storage to potentially persist through a glacial cycle (Harden et al., 1992). We thus suggest that the increased monsoonal precipitation and smaller ice volume during MIS 14 would have combined to increase land biomass that continued into MIS 13. The Northern Hemisphere thus had the potential to store light carbon in the terrestrial reservoir resulting in the enriched $\delta^{13}C$ MIS 13 signal seen in the ocean basins (Yin and Guo, 2008).

**4.2 Ocean circulation changes in the Atlantic basin**

One explanation for the glacial-interglacial variations in atmospheric $CO_2$ invokes a dominant role by the Southern Ocean in storing and releasing dissolved inorganic carbon (DIC) in the deep Southern Ocean, with deep-ocean sequestration of atmospheric $CO_2$ occurring through decreased upwelling and vertical mixing of AABW (Sigman et al., 2010). Expansion of Southern Ocean sea ice can also lower atmospheric $CO_2$ by insulating upwelled water from the atmosphere, thus reducing outgassing, and by increasing the volume of AABW and its capacity to hold DIC (Stephens and Keeling, 2000; Ferrari et al., 2014). According to this framework,

pre-MBT interglaciations with lower $CO_2$ would be associated with greater sea-ice extent and a larger volume of AABW, whereas post-MBT interglaciations with higher $CO_2$ suggest reduced sea-ice extent and AABW volume. Glacial values of $CO_2$ remain relatively constant throughout the last 800 kyr (Figure 1), suggesting that the change in relative AABW volume before and after the MBT only occurred during interglaciations.

This mechanism is consistent with ice-core evidence for greater sea-ice extent during pre-MBT interglaciations (Wolff et al., 2006) and with modeling results that show that interglacial AABW formation decreased after the MBT through insolation-induced feedbacks on sea ice and surface water density (Yin, 2013). Moreover, based on the Ba/Fe proxy of organic matter flux to the deep ocean south of the Polar Front, Jaccard et al. (2013) argued that the deep Southern Ocean reservoir was larger prior to the MBT.

Our analyses of changes in Atlantic $\delta^{13}C$ over the last 800 kyr further support an important role of AABW in causing the post-MBT increase in interglacial $CO_2$. In particular, the steeper latitudinal gradient between North and South Atlantic $\delta^{13}C$ records before the MBT reflects greater northward penetration of AABW, whereas the post-MBT decrease in gradient suggests greater southward penetration of NADW (Figure 10b). These gradient changes are further illustrated by contour plots of average interglacial $\delta^{13}C$ values in the Atlantic which show that prior to the MBT, AABW penetrated north of the equator, increasing the $\delta^{13}C$ gradient (Figure 12a), in contrast to remaining south of the equator after the MBT, decreasing the gradient (Figure 12c). Removal of MIS 13 and its associated enriched carbon isotope excursion further highlights the greater volume of AABW in the pre-MBT interglacial Atlantic (Figure 12b). We note that a record of the water mass tracer $\varepsilon_{Nd}$ from 6°N (Howe et al., 2017) is in good agreement with our North Atlantic regional $\delta^{13}C$ stack (Figure 10a), with both records suggesting that changes in volume of the interglacial AABW occurred south of the equator. This reorganization of the dominant interglacial water masses in the Atlantic basin across the MBT, perhaps resulting from insolation-induced feedbacks (Yin, 2013) would lead to a greater release of deep-ocean $CO_2$ during the post-MBT interglaciations, with corresponding warmer interglaciations (Figure 5). An alternative explanation for the observed decrease in latitudinal gradient could be changes in the isotopic composition of AABW across this time period. However, modeling results of long-term carbon fluctuations across this interval suggest that changes in the burial rate of organic and inorganic carbon caused the $\delta^{13}C$ depletion – the opposite signal necessary to create the increased similarity between northern- and southern-sourced waters (Hoogakker et al., 2006). Thus, it is more likely explained by changes in AABW influence north of the equator.

Cross-spectral analysis of pre-MBT North and South Atlantic $\delta^{13}C$ stacks indicates in-phase coherency between the records at the eccentricity and obliquity frequencies. Similar tests for the post-MBT $\delta^{13}C$ stacks exhibit coherency at eccentricity, obliquity, and precession frequencies, with the South Atlantic stack leading the North Atlantic by ~23º (7 kyr) in eccentricity, ~18º (2 kyr) in obliquity, and ~36º (2 kyr) in precession (Figure S6). All phase relationships overlap within uncertainty, suggesting that South Atlantic $\delta^{13}C$ leads North Atlantic $\delta^{13}C$ by 2-7 kyr following the MBT. This lead by the South Atlantic is most apparent during terminations (Figures 9, 12) and is most likely related to deglacial mechanisms for ventilation of respired $CO_2$ from the deep Southern Ocean such as enhanced wind-driven upwelling or the melting of sea ice in response to the bipolar seesaw (Cheng et al., 2009).

**5. Conclusions**

Using statistical analyses of multiple climate proxies, we have further characterized the Mid-BrunhesTransition as an increase in interglacial sea-surface and Antarctic temperatures, atmospheric $CO_2$, and $CH_4$ beginning with MIS 11. At the same time, our new analyses also document a number of changes in other components of the climate system that began as early as MIS 14 that suggest a more complex sequence of events prior to the MBT, although their relationship to the MBT remains unclear. Figure 13 highlights key features in the sequence of events beginning with an increase in Asian summer monsoon strength during MIS 15 that persisted through MIS 14 and into MIS 13. The strong monsoon strength during MIS 14 is associated with a weak glaciation, which in combination would have been conducive to a build-up of Northern Hemisphere land biomass. A continued strong Asian summer monsoon during MIS 13 associated with greater precipitation would have further sequestered land biomass and provided a reservoir for light carbon, resulting in the oceans becoming unusually enriched in $\delta^{13}C$ as recorded in the global benthic $\delta^{13}C$ carbon isotope excursion. MIS 12 was associated with the return of large ice sheets, collapse of the Asian summer monsoon, and the first increase in amplitude of Southern Hemisphere dust. A decrease in the latitudinal gradient of interglacial Atlantic $\delta^{13}C$ at the MBT suggests a reorganization of the water masses in the basin and reduction in the size of interglacial AABW, thus possibly explaining the increase in interglacial values of atmospheric $CO_2$ with corresponding increases in interglacial SSTs and $CH_4$. This evidence for a change in AABW is consistent with modeling results that suggest that the MBT was forced by insolation (Yin, 2013).

**7. Data availability**

All data related to this research, including input and analytical output data, are available at PANGAEA Data Publisher (https://doi.pangaea.de/10.1594/PANGAEA.894898; Barth et al., 2018). Additional data without persistent digital object

identifiers used in this study were retrieved from Bickert et al. (1997), Lisiecki et al. (2008), and Pisias et al. (1997) and are

345 available upon request.

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

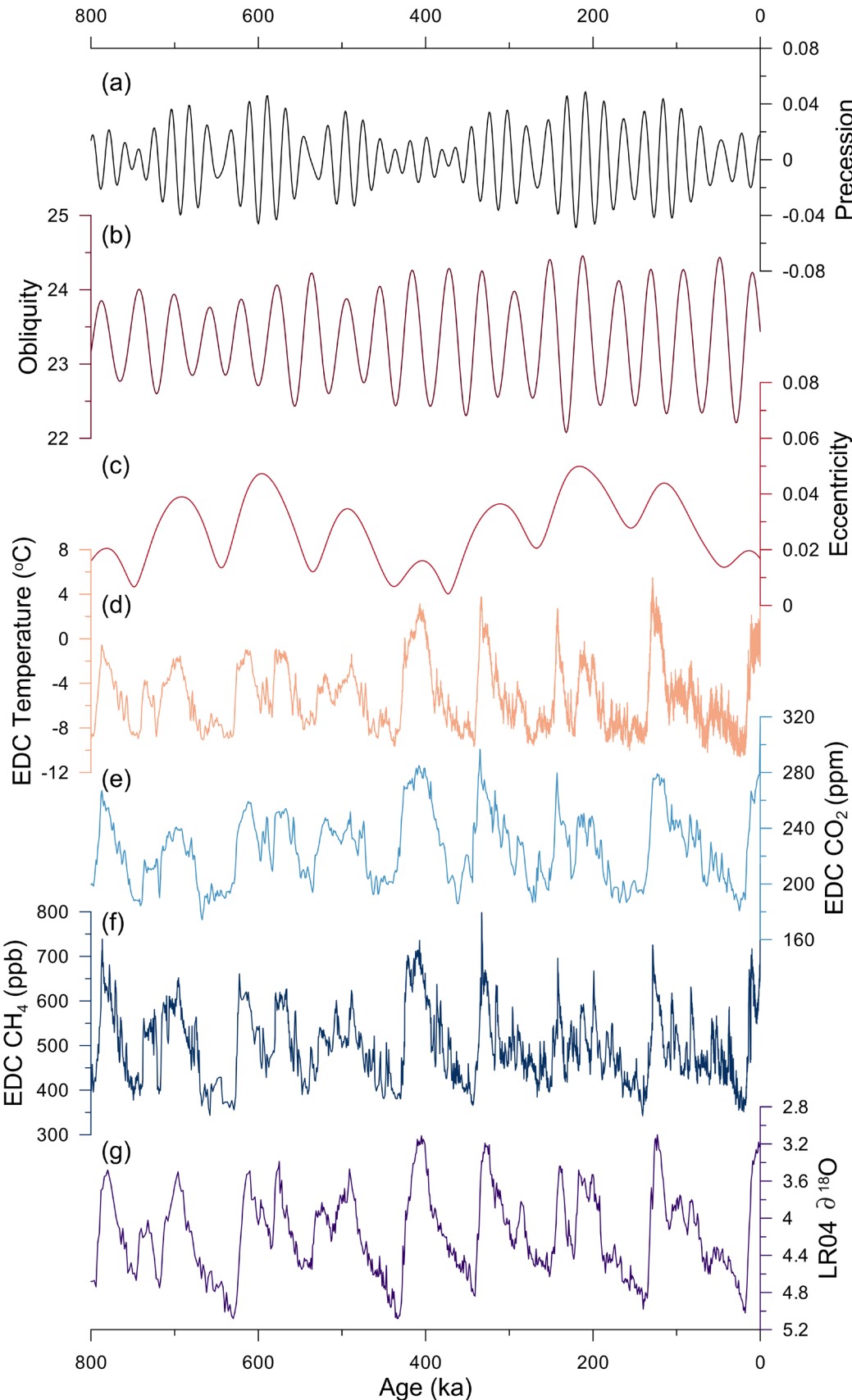

**Figure 1 – Orbital forcing and climate records for the last 800 kyr. a, b, c,** Precession, Obliquity, and Eccentricity (Laskar et al., 2004). **d,** Deuterium-derived temperature from the EPICA Dome C ice core in Antarctica (Jouzel et al., 2007). **e,** Atmospheric $CO_2$ from EPICA Dome C (EPICA community members, 2007, Lüthi et al., 2008). **f,** Atmospheric $CH_4$ from EPICA Dome C (EPICA community members, 2007). **g,** Global benthic oxygen isotope stack (Lisiecki and Raymo, 2005).

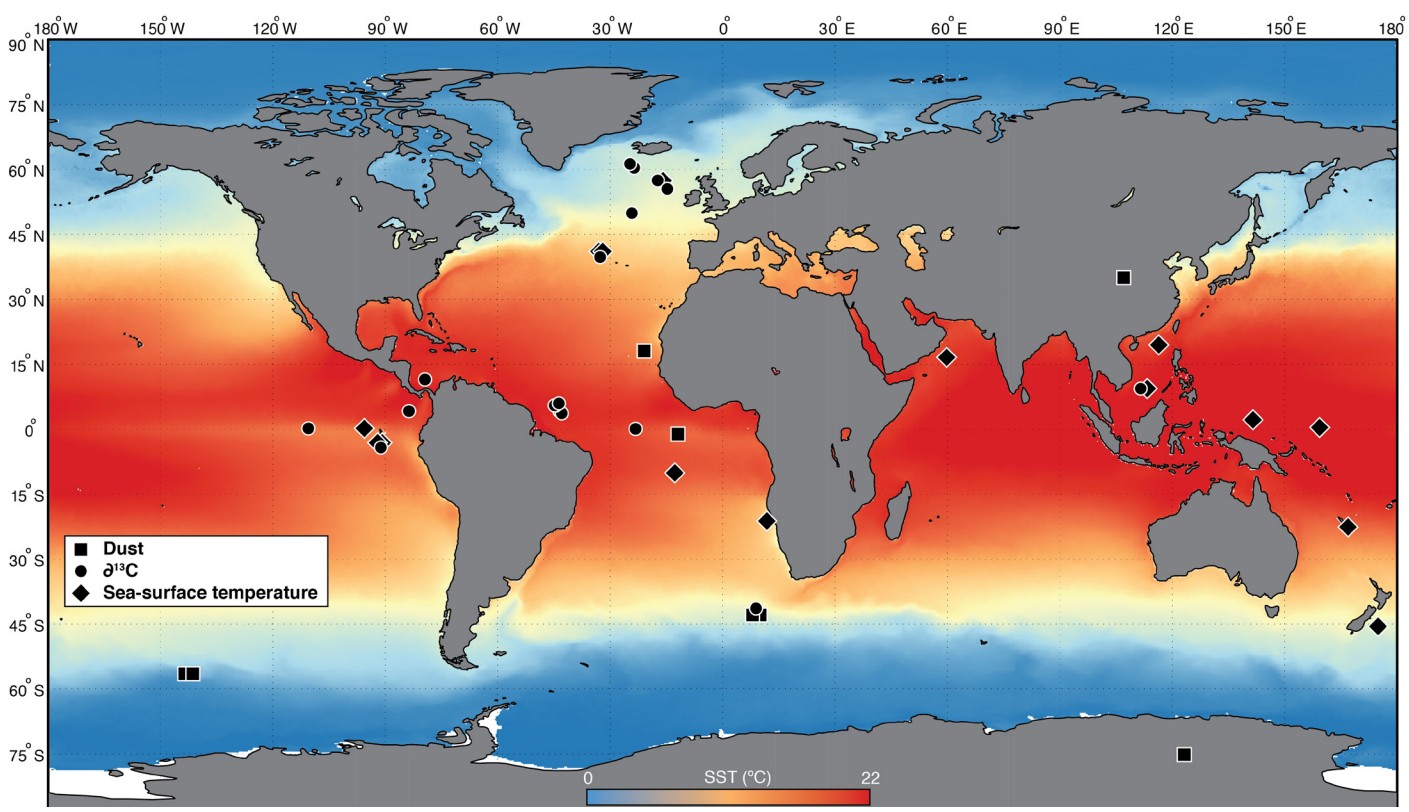

**Figure 2 – Site locations.** Map indicating the locations of the cores used in this research and modern sea-surface temperature values. Each symbol represents a different proxy record. Diamonds – sea-surface temperatures. Circles – benthic $\partial^{13}$C. Squares – dust.

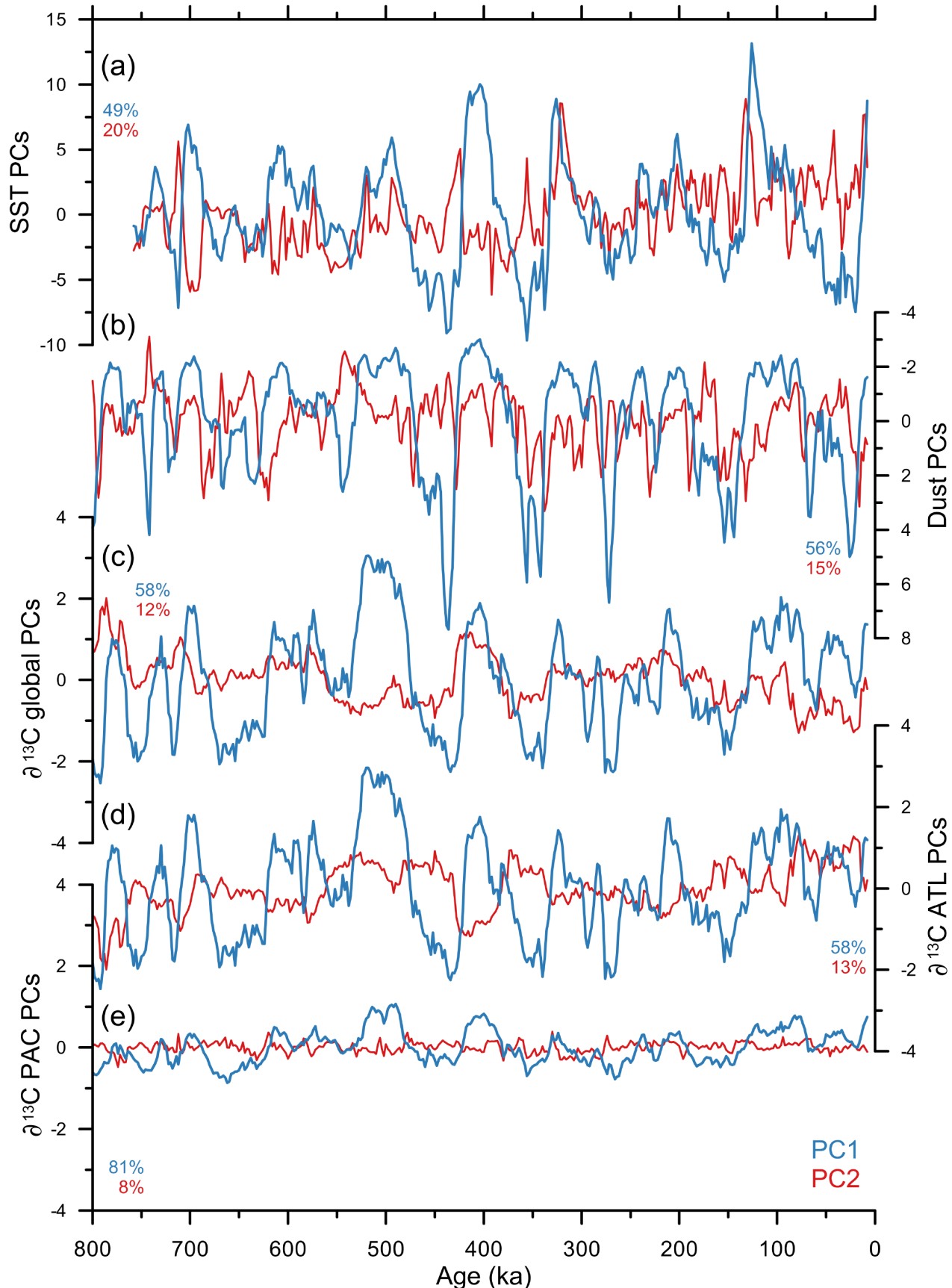

**Figure 3 – Principal components.** Plots of the first (PC1; blue) and second (PC2; red) principal components from our EOF analysis of each climate variable. Percent variance explained by each PC represented by the numbers with the corresponding color. **a,** Sea-surface temperatures. **b,** Dust records. **c,** Global $\partial^{13}C$. **d,** $\partial^{13}C$ of the Atlantic. **f,** $\partial^{13}C$ of the Pacific.

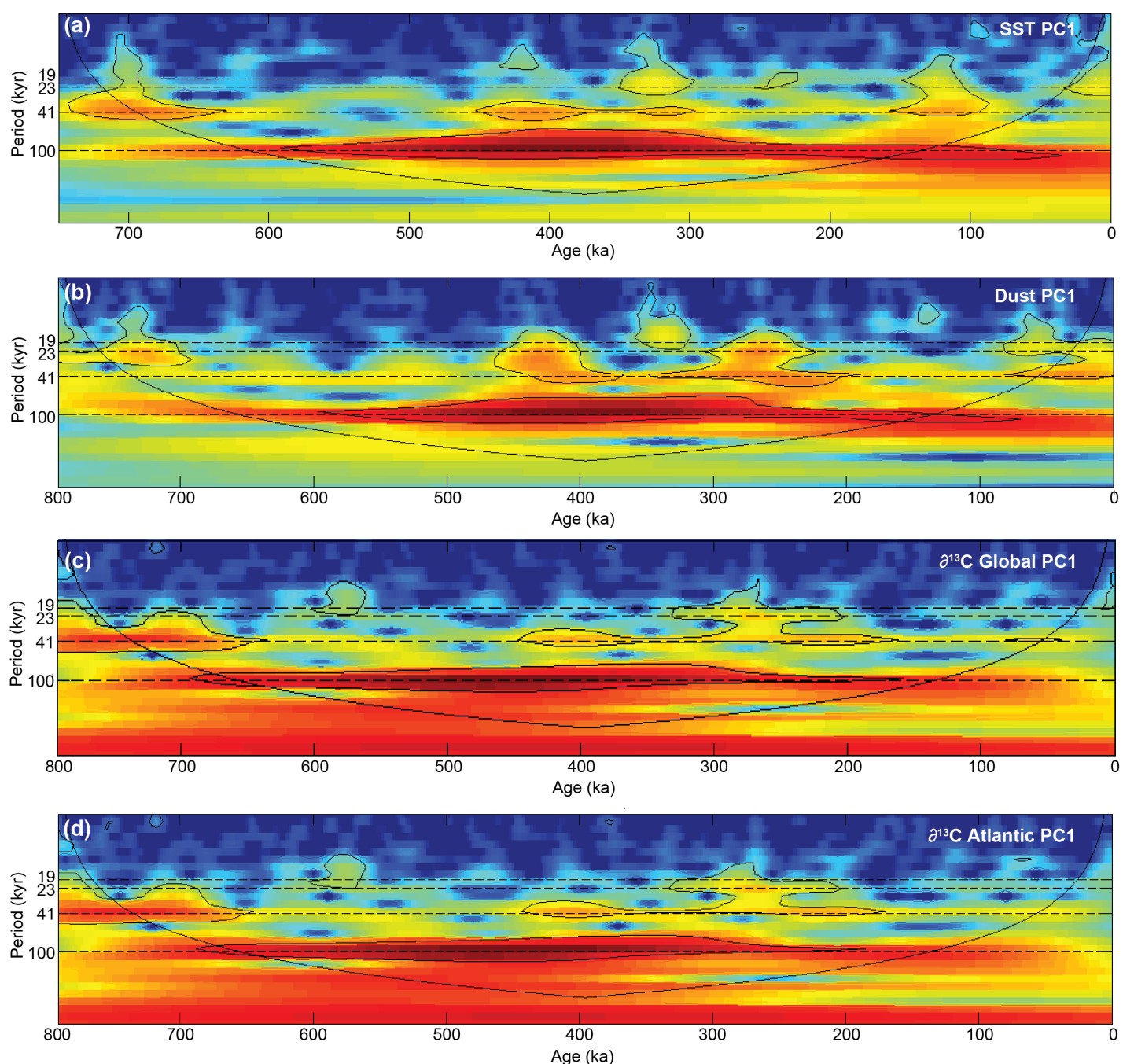

**Figure 4 – Wavelet analysis.** Wavelets of four of the first principal components. **a,** Sea-surface temperatures. **b,** Dust records. **c,** Global $\partial^{13}C$. **d,** $\partial^{13}C$ of the Atlantic. Red colors represent higher spectral power. Blue colors represent lower spectral power. Statistical significance highlighted by the thin black line. Milankovitch periods highlighted by the dashed horizontal lines.

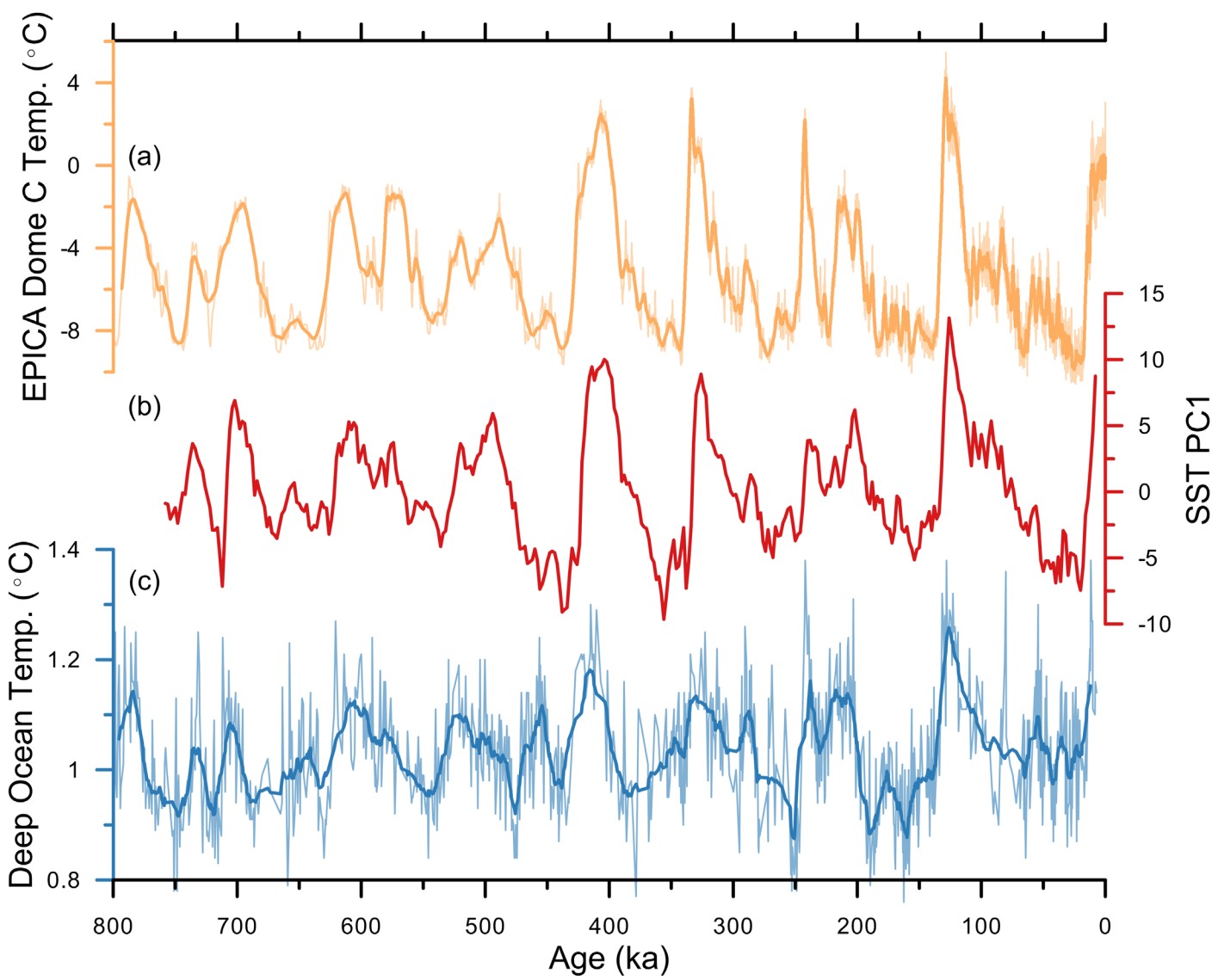

**Figure 5 – Temperature records. a,** Deuterium-based temperature record from EPICA Dome C in Antarctica (light yellow; Jouzel et al., 2007). The darker yellow line is a 15-point moving average. **b,** The first principal component of our sea-surface temperature analysis (red). **c,** Bottom water temperature derived from Mg/Ca measurements at ODP 1123 (light blue; Elderfield et al., 2012). Dark blue line is a 15-point moving average.

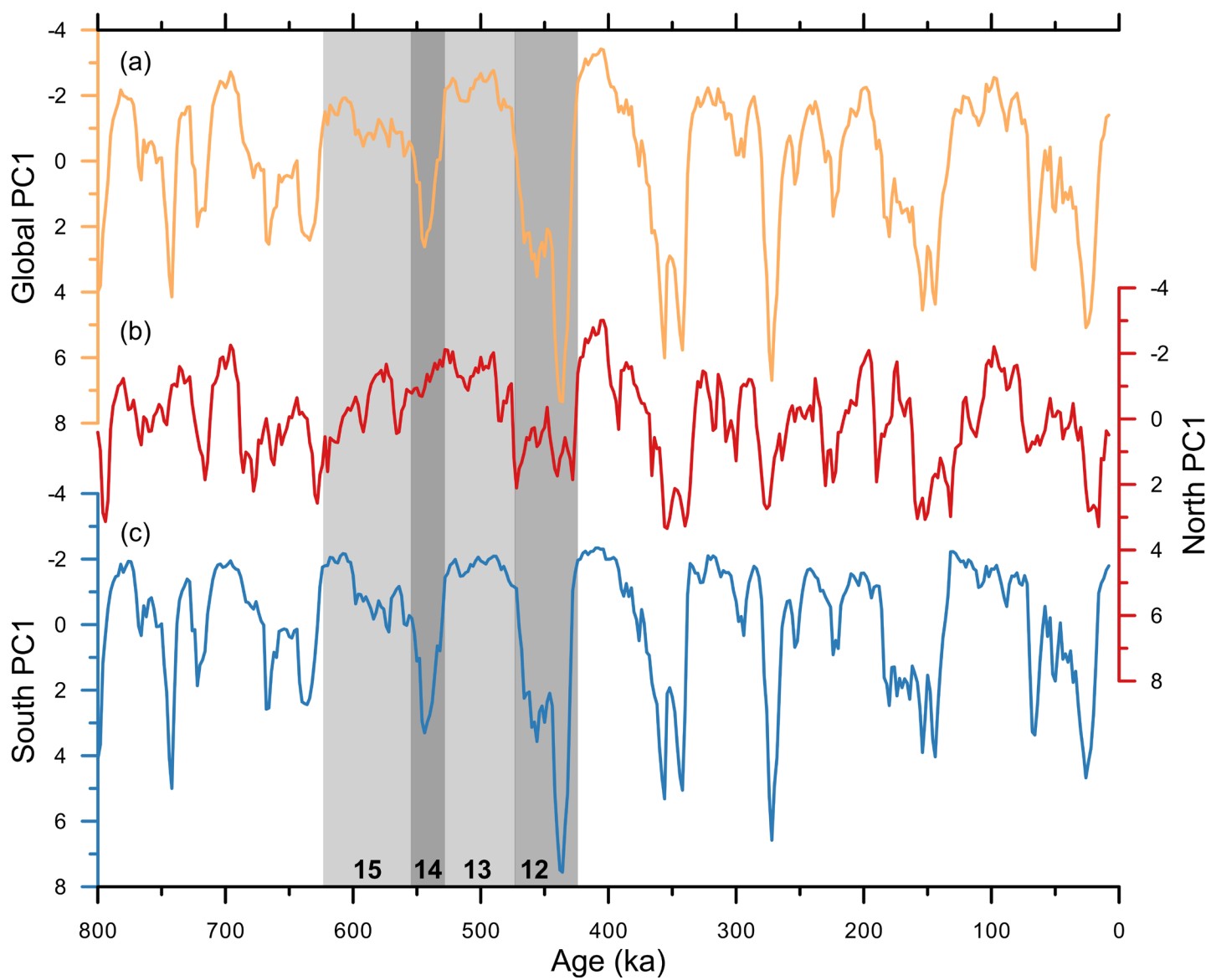

**Figure 6 – Dust principal components.** The first principal components of our dust analysis for the global (yellow), north (red), and south (blue) records. Vertical gray boxes highlight specific glacial (dark gray) and interglacial (light gray) periods. The numbers indicate the associated Marine Isotope Stage of each box.

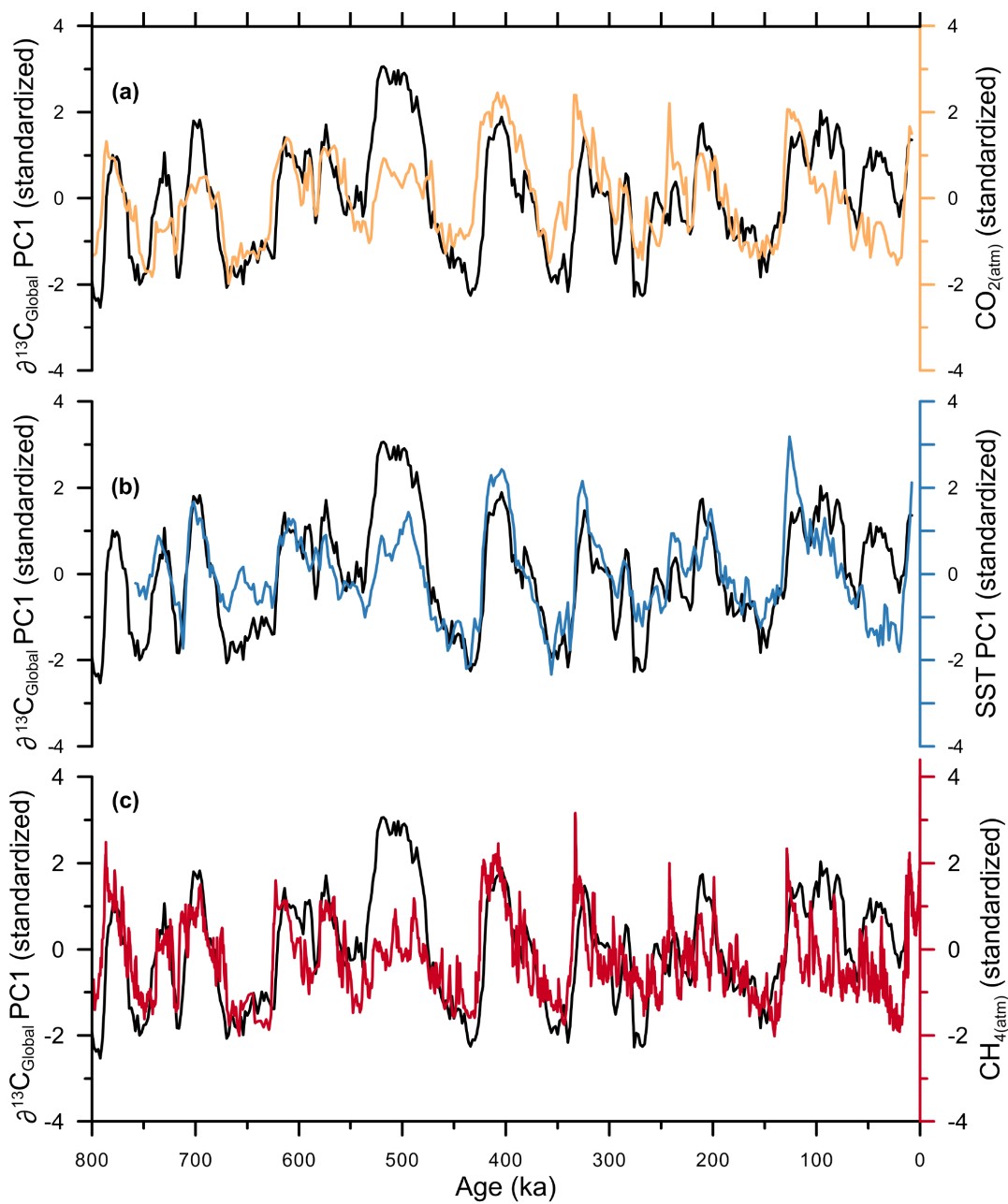

**Figure 7 – Global ∂¹³C proxy comparison.** Comparison of the global ∂¹³C first principal component (PC1; black) compared against **a,** EPICA Dome C $CO_2$ (yellow; EPICA community members, 27, Lthi et al., 28), **b,** sea-surface temperature PC1 from this research (blue), and **c,** EPICA Dome C $CH_4$ (red; EPICA community members, 2007).

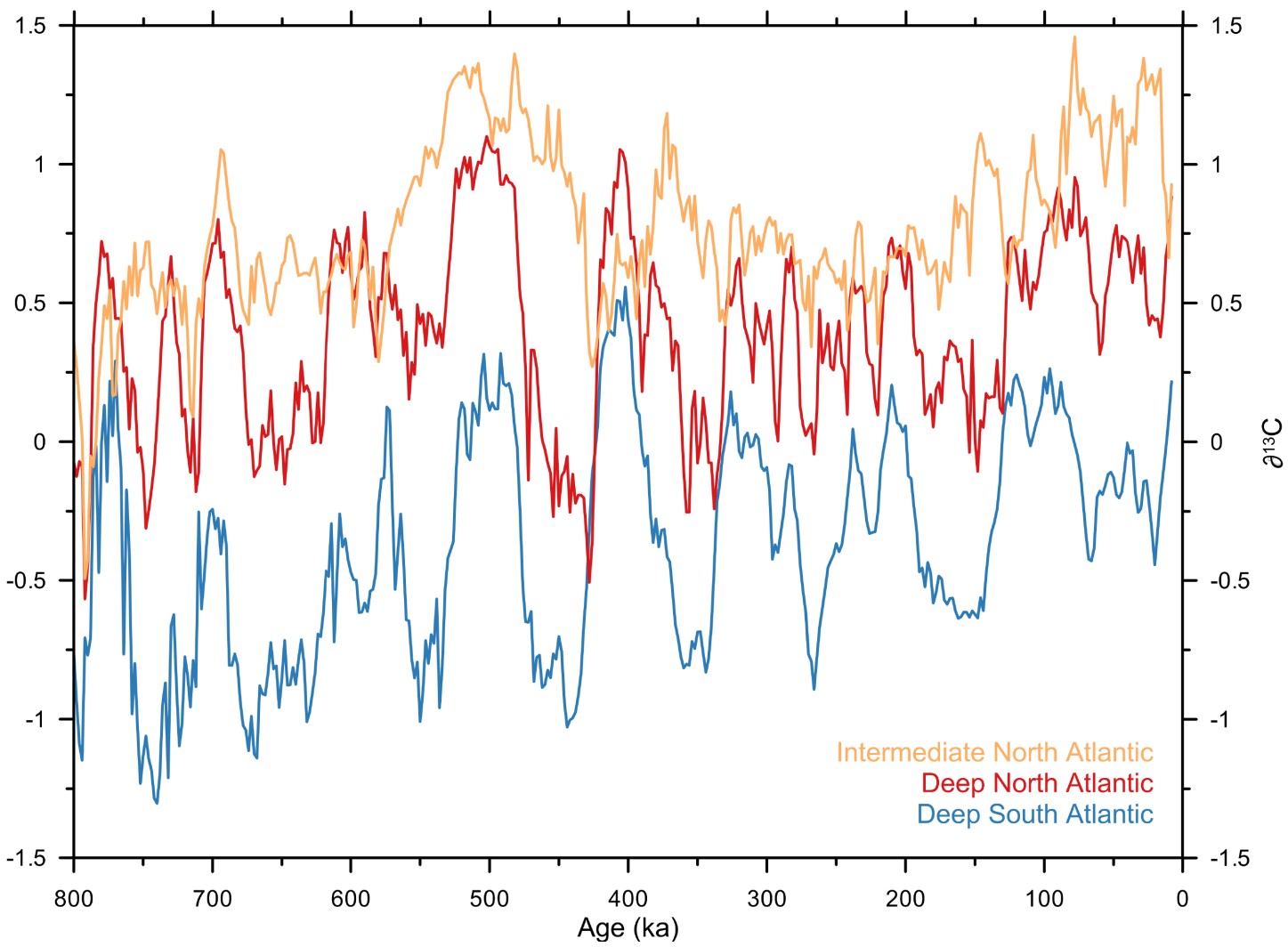

**Figure 8 – Regional $\partial^{13}C$ stacks.** Stacked records of benthic $\partial^{13}C$ separated into three regions: Intermediate North Atlantic (orange), Deep North Atlantic (red), and Deep South Atlantic (blue). All plots shown in $\partial^{13}C$ space to highlight different isotopic values.

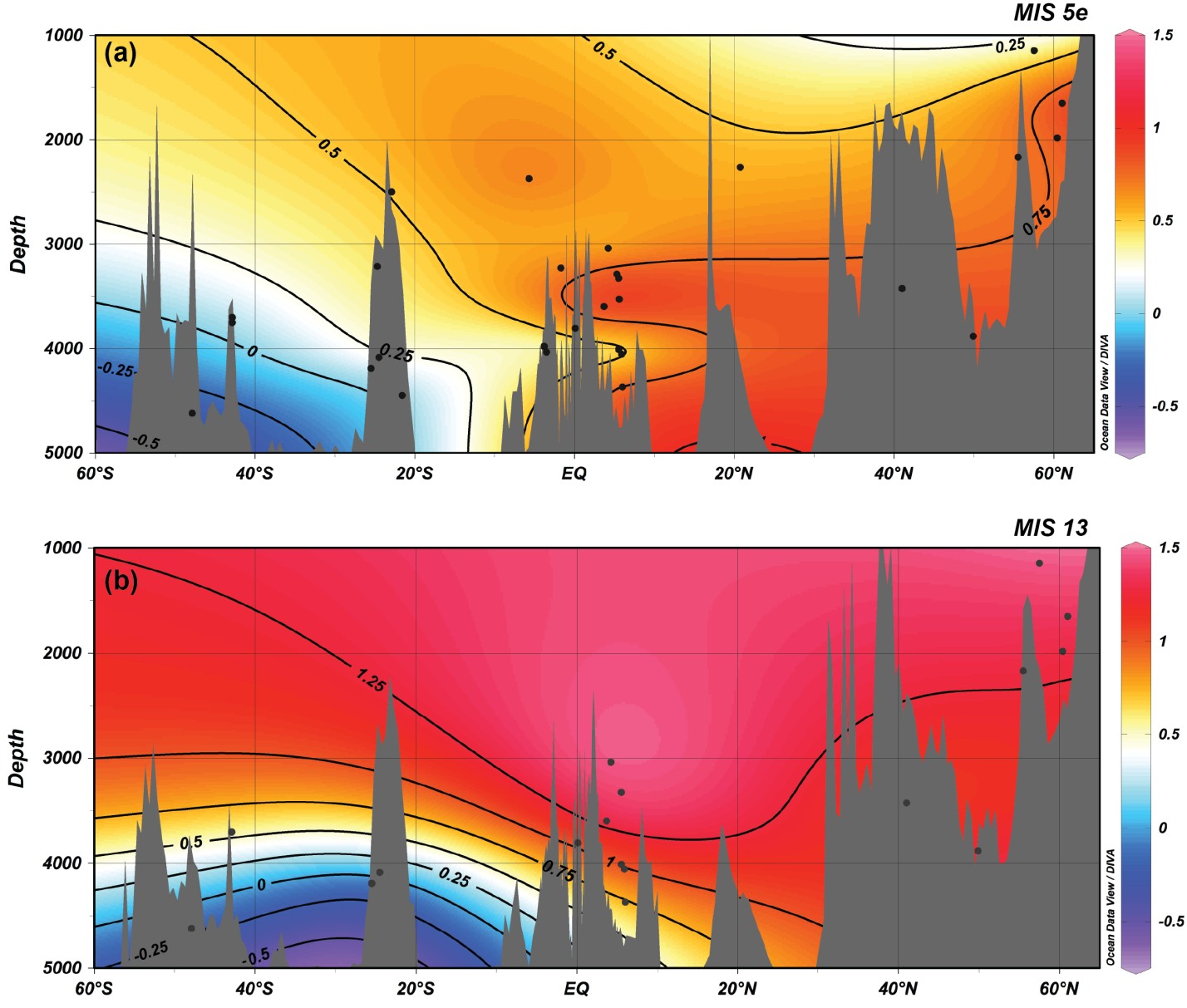

**Figure 9 – MIS 13 and 5e contour plots of $\partial^{13}C$.** Contour plots of the $\partial^{13}C$ values in the North Atlantic basin for the interglacials MIS 13 and MIS 5e. Red colors represent more positive, enriched values. Blue colors represent lower, depleted values. Plot created using Ocean Data Viewer.

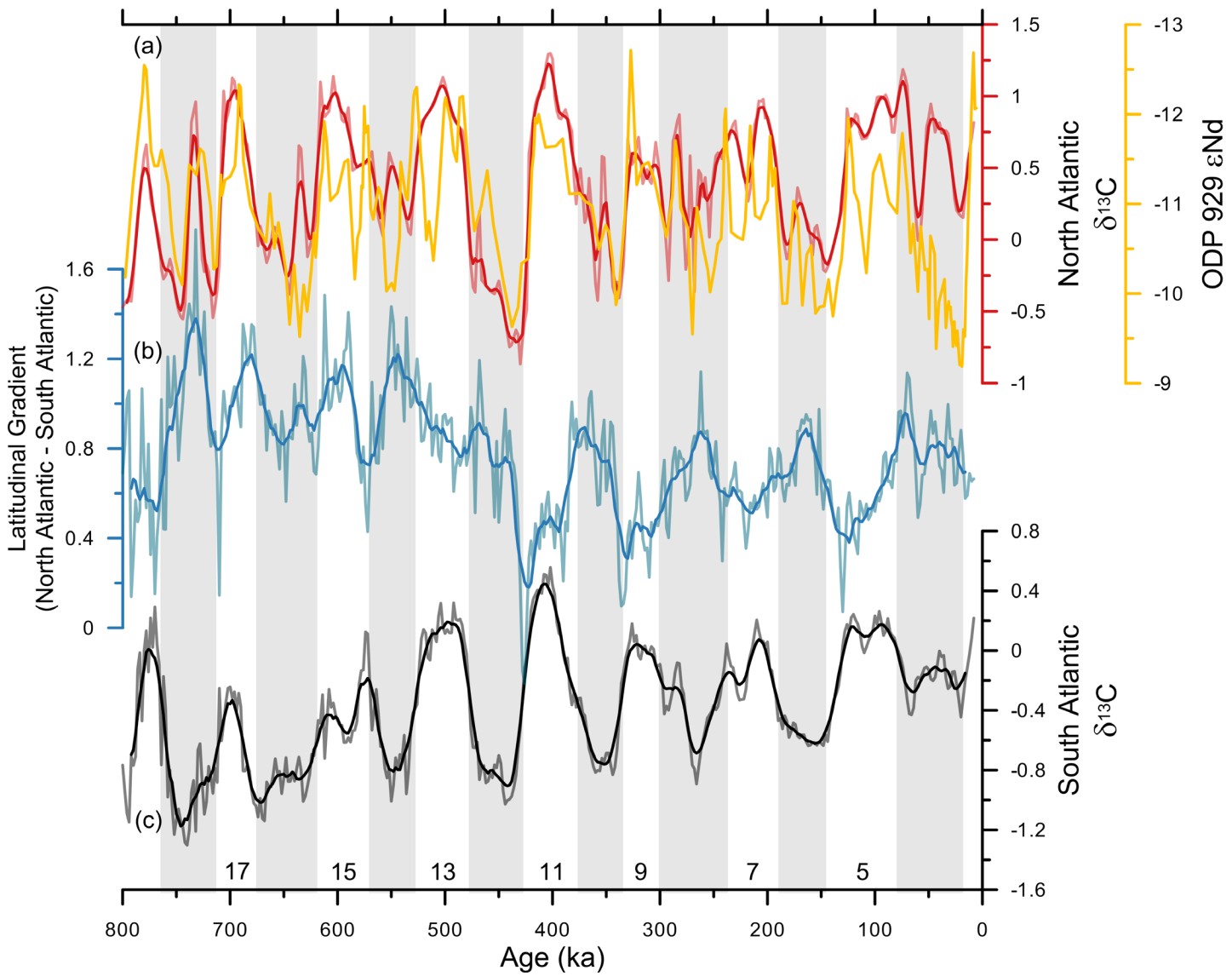

**Figure 10 – Latitudinal $\partial^{13}C$ gradient. a,** North Atlantic regional $\partial^{13}C$ stack plotted in $\partial^{13}C$ space (red) authigenic εNd (yellow; Howe et al., 2017). **b,** Latitudinal gradient of Atlantic $\partial^{13}C$ regional stacks (North Atlantic minus South Atlantic; blue). Lower values demonstrate increased similarity between the records. **c,** South Atlantic regional $\partial^{13}C$ stack plotted in $\partial^{13}C$ space (black). Vertical gray bars indicate glacial periods. Numbers represent Marine Isotope Stage numbers for interglacials.

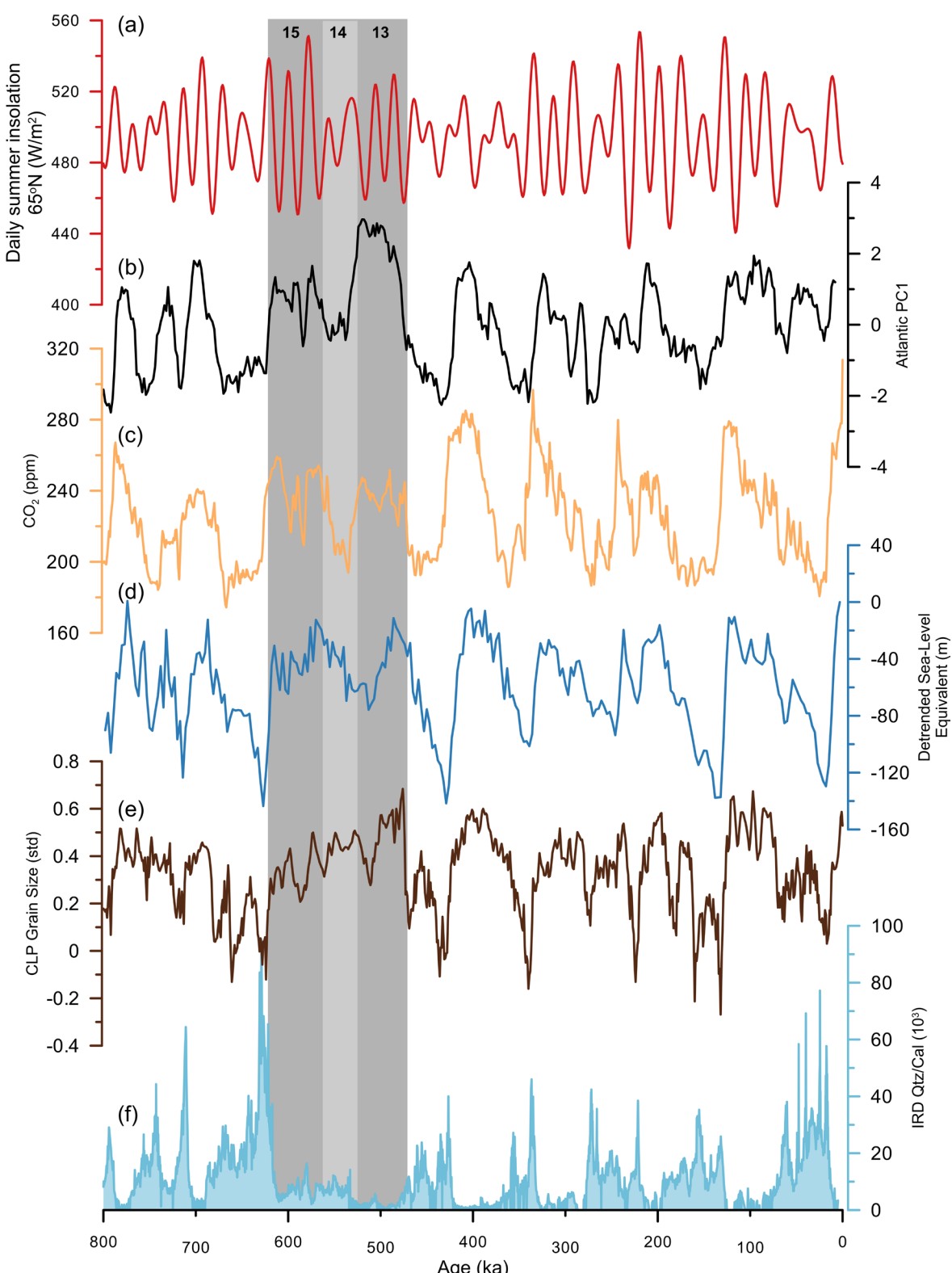

**Figure 11** – Marine isotope stages 15 to 13 and the carbon isotope excursion. **a,** Summer insolation at 65° N (red; Laskar et al., 2004). **b,** First principal component of Atlantic $\partial^{13}C$ (black). **c,** EPICA Dome C $CO_2$ (yellow; EPICA community members, 2004, Lüthi et al., 2008). **d,** Detrended sea-level equivalent from Shakun et al., 2015 (blue). Derived from $\partial^{18}O_{sw}$ calculations. Negative numbers indicate lower sea level and increased ice volume. **e,** Chinese Loess Plateau grain size indicating relative Asian summer monsoon strength (brown; Sun et al., 25). **f,** Quartz/Calcite ratios from site U1313 in the North Atlantic as a measure of ice-rafted debris (light blue; Naafs et al., 212). Dark gray bars highlight the interglacials (MIS 15 and MIS 13) between ~630 to ~470 ka. Light gray bar highlights MIS 14.

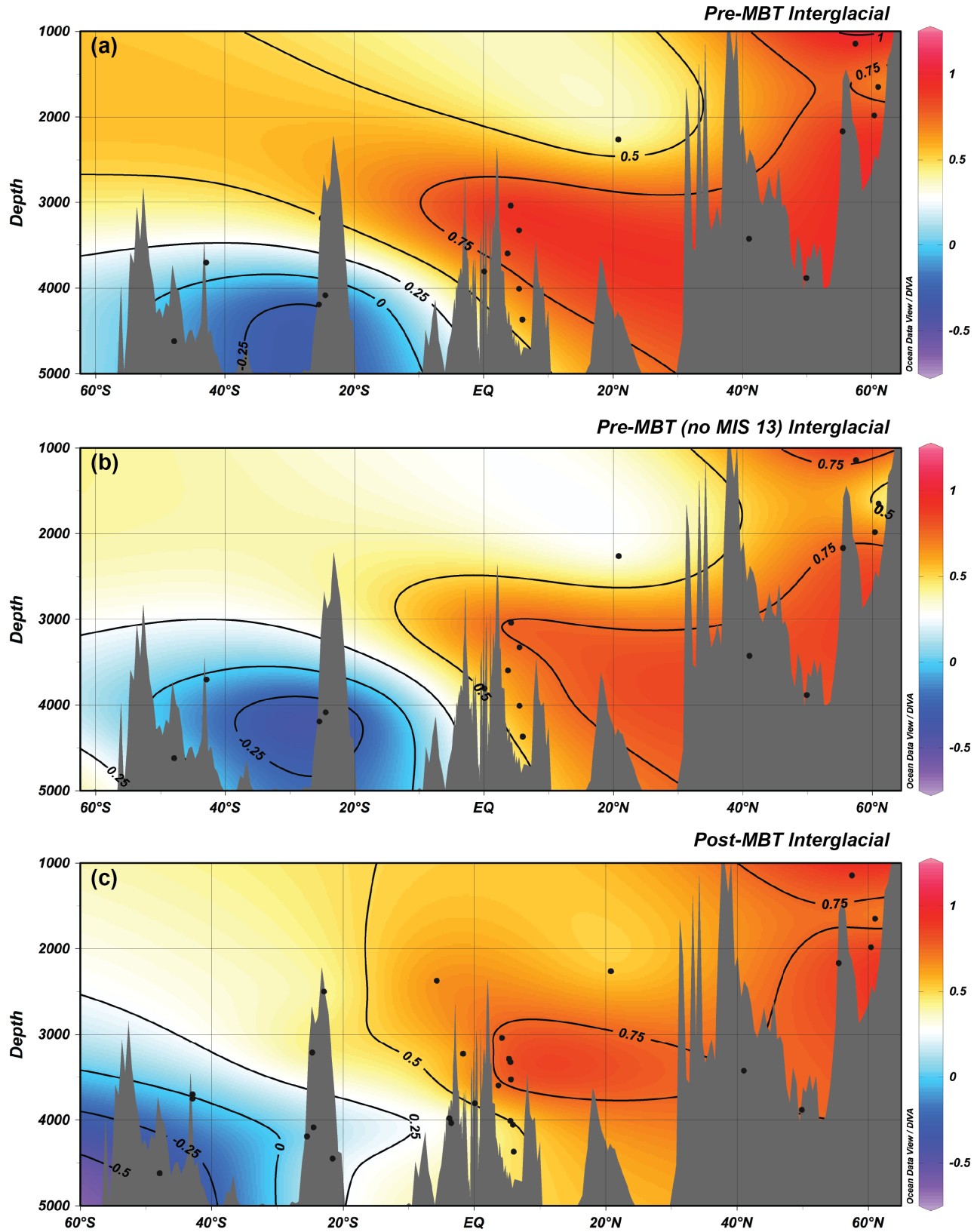

**Figure 12 – Average interglacial $\partial^{13}C$ contours.** Contour plots of the average interglacial $\partial^{13}C$ values in the Atlantic for **a,** pre-MBT included MIS 13, **b,** pre-MBT excluding MIS 13 (enriched carbon isotope excursion), and **c,** post-MBT. Red colors indicate higher $\partial^{13}C$ values. Blue colors indicate lower $\partial^{13}C$ values. Boundary between the two water masses (NADW and AABW) indicated at the 0.25‰ contour (Curry and Oppo, 2005).

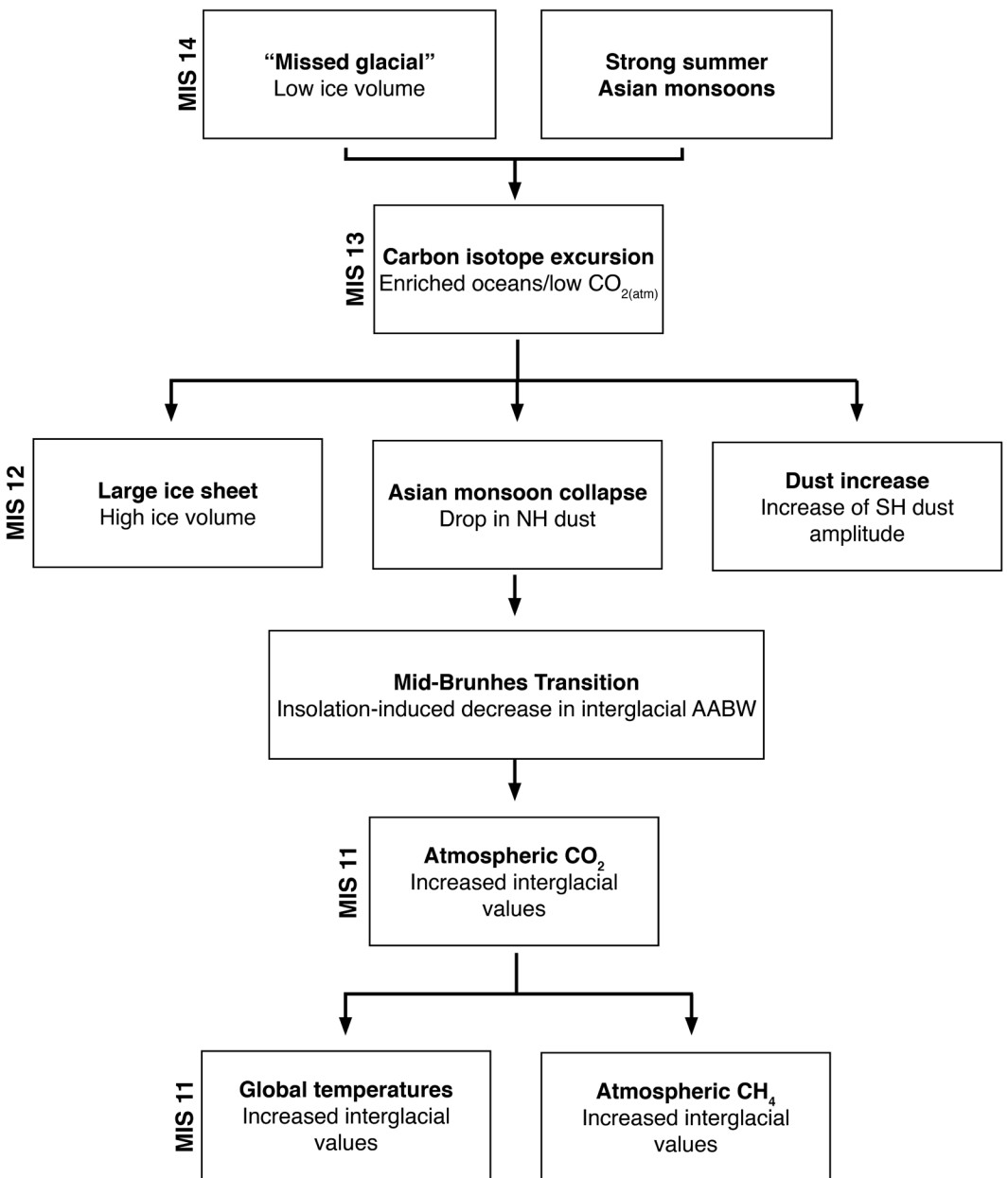

**Figure 13 – Schematic representation of the sequence of events leading to the Mid-Brunhes Transition.** Corresponding marine isotope stages are located on the left side of each row. Boxes in a row indicate synchronous events.

*Table 1 - Data compilation*

| Core | Location | Latitude | Longitude | Δt (ka) | Proxy | doi/url | Reference |
|---|---|---|---|---|---|---|---|
| DSDP 607 | Northeastern Atlantic | 41.0012 | -32.9573 | 2.8 | SST - Transfer function | https://doi.org/10.1594/PANGAEA.701229 | Ruddiman et al., 1989 |
| ODP 846 | Eastern Equatorial Pacific | -3.0949 | -90.818 | 2.3 | SST - Uk37 | https://doi.org/10.1126/science.1185435 | Herbert et al., 2010 |
| ODP 982 | North Atlantic | 57.5165 | -15.8667 | 4.7 | SST - Uk37 | https://www.ncdc.noaa.gov/paleo/study/8624 | Lawrence et al., 2009 |
| ODP 1143 | Western Equatorial Pacific | 9.3619 | 113.2851 | 1.9 | SST - Uk37 | https://doi.org/10.1594/PANGAEA.786444 | Li et al., 2011 |
| ODP 1082 | South Atlantic | -21.0941 | 11.8205 | 4.5 | SST - Uk37 | https://doi.org/10.1594/PANGAEA.786701 | Etourneau et al., 2009 |
| ODP 1313 | North Atlantic | 41.0011 | -32.9573 | 1.4 | SST - Uk37 | https://doi.org/10.1594/PANGAEA.744483 | Naafs et al., 2012 |
| ODP 722 | Arabian Sea | 16.6218 | 59.7953 | 2.0 | SST - Uk37 | https://doi.org/10.1126/science.1185435 | Herbert et al., 2010 |
| ODP 1146 | South China Sea | 19.4567 | 116.2727 | 1.6 | SST - Uk37 | https://doi.org/10.1126/science.1185435 | Herbert et al., 2010 |
| ODP 846 | Eastern Equatorial Pacific | -3.0949 | -90.818 | 1.3 | SST - Uk37 | https://doi.org/10.1038/nature02338 | Liu and Herbert, 2004 |
| MD97-2140 | Western Pacific Warm Pool | 2.1792 | 141.4581 | 3.9 | SST - Mg/Ca | https://www.ncdc.noaa.gov/paleo/study/6266 | de Garidel-Thoron, 2005 |
| MD06-3018 | Tropical Western Pacific | -22.5977 | 166.8628 | 5.2 | SST - Mg/Ca | https://www.ncdc.noaa.gov/paleo/study/11188 | Russon et al., 2011 |
| ODP 806 | Western Equatorial Pacific | 0.319 | 159.361 | 2.4 | SST - Mg/Ca | https://doi.org/10.1594/PANGAEA.772015 | Medina-Elizalde and Lea, 2005 |
| DSDP 594 | Southwest Pacific | -45.5235 | 174.948 | 2.6 | SST - Modern analog | https://doi.org/10.1594/PANGAEA.691478 | Schaefer et al., 2005 |
| V22-174 | South Atlantic | -10.0667 | -12.8167 | 4.1 | SST - Transfer function | https://doi.org/10.1594/PANGAEA.52228 | Specmap, 1990 |
| RC13-110 | Eastern Equatorial Pacific | 0 | -96 | 4.9 | SST - Transfer function | | Pisias et al., 1997 |
| ODP 659 | Eastern Equatorial Atlantic | 18.0772 | -21.0262 | 3.6 | Dust flux | https://doi.org/10.1594/PANGAEA.696121 | Tiedemann et al., 1994 |
| ODP 1090 | Subantarctic Atlantic | -42.9137 | 8.8997 | 0.3 | Dust MAR | https://doi.org/10.1594/PANGAEA.767460 | Martinez-Garcia et al., 2011 |
| ODP 1090 | Subantarctic Atlantic | -42.9137 | 8.8997 | 0.3 | Fe MAR | https://doi.org/10.1594/PANGAEA.767460 | Martinez-Garcia et al., 2011 |
| CLP | Chinese Loess Plateau | | | 1.0 | Grain size | https://doi.org/10.1029/2006GC001287 | Sun et al., 2005 |
| Lake Baikal | Southern Russia | | | 0.5 | Silica % | https://www.ncdc.noaa.gov/paleo/study/6068 | Prokopenko et al., 2006 |
| PS75-074 | Pacific Southern Ocean | -56.4696 | -142.9954 | 0.4 | Fe counts | https://doi.org/10.1594/PANGAEA.826600 | Lamy et al., 2014 |
| PS75-076 | Pacific Southern Ocean | -56.4696 | -142.9954 | 0.4 | Fe wt. % | https://doi.org/10.1594/PANGAEA.826600 | Lamy et al., 2014 |
| ODP 663 | Eastern Equatorial Atlantic | -1.1978 | -11.8785 | 2.7 | Terrestrial % | https://doi.org/10.1594/PANGAEA.208129 | deMenocal et al., 1993 |
| EPICA Dome C | Antarctica | -75.06 | 123.21 | 0.2 | Dust | https://doi.org/10.1594/PANGAEA.695995 | Lambert et al., 2008 |
| ODP 982 | North Atlantic | 57.5 | -15.9 | 2.5 | $\partial^{13}C$ | https://doi.org/10.1594/PANGAEA.700897 | Venz et al., 1999 |
| ODP 983 | North Atlantic | 60.4 | -23.6 | 1.0 | $\partial^{13}C$ | https://www.ncdc.noaa.gov/paleo/study/2543 | McIntyre et al., 1999 |
| ODP 984 | North Atlantic | 61 | -24 | 3.5 | $\partial^{13}C$ | https://www.ncdc.noaa.gov/paleo/study/5897 | Raymo et al., 2004 |
| DSDP 607 | North Atlantic | 41 | -33 | 4.1 | $\partial^{13}C$ | https://doi.org/10.1594/PANGAEA.52379 | Ruddiman et al., 1989 |
| ODP 658 | North Atlantic | 20.8 | -18.7 | 1.6 | $\partial^{13}C$ | https://doi.org/10.1594/PANGAEA.68570 | Tiedemann et al., 1994 |
| U 1308 | North Atlantic | 49.9 | -24.2 | 0.3 | $\partial^{13}C$ | https://www.ncdc.noaa.gov/paleo/study/10250 | Hodell et al., 2008 |
| ODP 980/981 | North Atlantic | 55.5 | -14.7 | 1.6 | $\partial^{13}C$ | https://doi.org/10.1594/PANGAEA.698998 | Oppo et al., 1998 |
| DSDP 502 | Equatorial Atlantic | 11.5 | -79.4 | | $\partial^{13}C$ | https://doi.org/10.1594/PANGAEA.701470 | deMenocal et al. 1992 |
| ODP 664 | Equatorial Atlantic | 0.1 | -23.2 | 3.2 | $\partial^{13}C$ | https://www.ncdc.noaa.gov/paleo/study/2529 | Raymo et al., 1997 |
| ODP 925 | Equatorial Atlantic | 4.2 | -43.5 | 4.3 | $\partial^{13}C$ | | Bickert et al., 1997 |
| ODP 926 | Equatorial Atlantic | 3.7 | -42.9 | 2.7 | $\partial^{13}C$ | | Lisiecki et al., 2008 |
| ODP 927 | Equatorial Atlantic | 5.5 | -44.5 | | $\partial^{13}C$ | | Bickert et al., 1997 |
| ODP 929 | Equatorial Atlantic | 5.5 | -44.5 | 4.9 | $\partial^{13}C$ | | Bickert et al., 1997 |
| ODP 928 | Equatorial Atlantic | 5.5 | -44.8 | 2.5 | $\partial^{13}C$ | | Lisiecki et al., 2008 |
| ODP 1090 | South Atlantic | -42.9 | 8.9 | 2.8 | $\partial^{13}C$ | https://doi.org/10.1594/PANGAEA.696106 | Venz and Hodell, 2002 |
| GeoB 1032 | South Atlantic | -22.9 | 6 | 3.7 | $\partial^{13}C$ | https://doi.org/10.1594/PANGAEA.54655 | Wefer et al., 1996 |
| GeoB 1035 | South Atlantic | -21.6 | 5 | 3.9 | $\partial^{13}C$ | https://doi.org/10.1594/PANGAEA.58766 | Bickert and Wefer, 1996 |
| ODP 1089 | South Atlantic | -47.9 | 9.9 | 0.4 | $\partial^{13}C$ | https://doi.org/10.1594/PANGAEA.701432 | Hodell et al., 2003 |
| GeoB 1211 | South Atlantic | -24.5 | 7.5 | 4.9 | $\partial^{13}C$ | https://doi.org/10.1594/PANGAEA.103634 | Bickert and Wefer, 1996 |
| GeoB 1214 | South Atlantic | -24.7 | 7.2 | 4.5 | $\partial^{13}C$ | https://doi.org/10.1594/PANGAEA.103635 | Bickert and Wefer, 1996 |
| RC13-229 | South Atlantic | -25.5 | 11.3 | 3.8 | $\partial^{13}C$ | https://doi.org/10.1594/PANGAEA.701361 | Oppo et al., 1990 |
| TN 576 | South Atlantic | -42.9 | 8.9 | 1.5 | $\partial^{13}C$ | https://www.ncdc.noaa.gov/paleo/study/2576 | Hodell et al., 2000 |
| ODP 1143 | Pacific | 9.4 | -246.7 | 3.8 | $\partial^{13}C$ | https://doi.org/10.1594/PANGAEA.784150 | Cheng et al., 2004 |
| ODP 677 | Pacific | 4.2 | -83.7 | 2.8 | $\partial^{13}C$ | https://doi.org/10.1594/PANGAEA.701316 | Shackleton et al., 1990 |
| ODP 846 | Pacific | -3.1 | -90.8 | 2.2 | $\partial^{13}C$ | https://doi.org/10.1594/PANGAEA.808207 | Mix et al., 1995 |
| ODP 849 | Pacific | 0.2 | -110.5 | 3.5 | $\partial^{13}C$ | https://doi.org/10.1594/PANGAEA.701400 | Mix et al., 1995 |
| ODP 1123 | Southwest Pacific | -41.7862 | -171.499 | 0.8 | Mg/Ca | https://doi.org/10.1594/PANGAEA.786205 | Elderfield et al., 2012 |
| EPICA Dome C | Antarctica | -75.06 | 123.21 | 0.4 | $CH_4$ | https://www.ncdc.noaa.gov/paleo/study/6093 | Loulergue et al., 2008 |
| EPICA Dome C | Antarctica | -75.06 | 123.21 | 0.4 | $CO_2$ | https://www.ncdc.noaa.gov/paleo/study/6091 | Luthi et al., 2008 |
| EPICA Dome C | Antarctica | -75.06 | 123.21 | 3.0 | Deuterium | https://www.ncdc.noaa.gov/paleo/study/6080 | EPICA Community Members, 2004 |