# Peer review of "Climate evolution across the Mid-Brunhes Transition"

_Climate of the Past, 2018_

## Short Comment (SC1) · 29 Apr 2018

The PAGES Data Stewardship Integrative Activity seeks to advance best practices for sharing the data generated and assembled as part of all PAGES-related activities. The CP Special Issue, "PAGES Young Scientists Meeting 2017" is part of this PAGES activity. The co-editors of the Special Issue are reviewing the data availability within each of the CP-Discussion papers in relation to the CP data policy (https://www.climate-of-thepast.net/about/data_policy.html) and current best practices. The editor team is making recommendations for each paper, with the goal of achieving a high and consistent level of data stewardship across the Special Issue. We recognize that an additional effort will likely be required to meet the high level of data stewardship envisaged, and we appreciate the dedication and contribution of the authors. This includes the use of

Data Citations (see example below). Authors are also strongly encouraged to deposit significant code into a suitable repository and to cite it using a Data Citation.

We ask authors to respond to our comments as part of the regular open interactive discussion. If you have any questions about PAGES Data Stewardship principles, please contact any of us directly.

Best wishes for the success of your paper.

YSM Special Issue editor team

R.L. Barnett, D.S. Kaufman, M.F. Loutre, M.N. Evans, S.C. Fritz, C. Tabor, Y. Zhang, H. Plumpton, E. Razanatsoa, and E. Dearing Crampton Flood

For this paper:

All papers submitted to "Climate of the Past" must include a Data Availability section that details the location of the data that were used as input to the study, including previously published data that were used for comparison purposes, and the data that were generated by the study.

Input Data (used for analyses) – Data citations are needed for all time-series data that were used in this study. Add a column to Table S1 to list the persistent identifier (doi or URL from NOAA-Paleo) for each record. We recommend that Table S1 be moved to the main text so that data generators are acknowledged more prominently. Change the "researcher" column in Table S1 to "reference" and include the bibliographic citation in the References. If the data are not already available in long-standing data repositories, then we ask the authors to work with the data generators (if possible) to rescue the data and transfer them along with essential metadata to a data repository (NOAA-Paleo, PANGEA, or other registered at re3data.org). Once curated in a pubic repository, each dataset will be assigned a persistent identifier, which is then cited in this paper to credit the original data generators, in addition to the bibliographic reference.

In this study, the input data were modified for some records to adjust the age scale,

and all time series were interpolated at 2-kyr intervals. To enable others to reproduce the primary results of this study, and to avoid losing this excellent compilation of proxy data, please transfer this entire modified dataset to a public repository, along with basic metadata for each record. The compilation should include the data and bibliographic citations for each of the constituent records, and the package (with its own persistent identifier) should be attributed to the authors of this study as an outcome of this synthesis.

Input Data (used for comparison) - Please cite and reference the source of the insolation curves shown in Figs. 1a,b,c, 14a. Because these data are used to compare rather than generate the primary outcome comes of the study, we suggest that a bibliographic reference to the data would suffice, although the authors are encouraged to include a data citation in addition. If, however, the time series were generated by the authors using online software, then both the data behind the calculator and the developer of the software should be cited.

Output Data (analytical) – Please prepare and upload to a registered data repository all of the data and metadata resulting from the primary analyses in the manuscript. This includes the time series of record stacks (Fig. 9), and derived temperature gradients (Fig. 12). Once assigned a persistent identifier (doi or URL from NOAA-Paleo), include the data citation for these files in the manuscript. Most data repositories use landing pages that collect the individual data files under a single identifier.

Output Data (statistical) – For the results of the statistical analyses including PCs (Fig. 4, 6b, 7, 8, 13a) spectral analyses (Figs. 3, 5) and d13C contour plots (Figs. 11, 15), please ensure that the statistical packages that were used to produce the plots are cited and fully referenced in the manuscript with sufficient details so that the plots could be replicated in future. We encourage the authors to upload the results of the statistical analyses to a repository (as above), but leave it to them to determine the likely future utility of the digital result of the statistical analyses relative to the effort involved to curate them.

The editorial team for the PAGES YSM special issue will review the datasets and data citations prior to accepting the manuscript. As such, please provide them as part of the author responses to editor and reviewer comments. In addition, please reply directly to this data-review comment using the Interactive Discussion so that this exchange can be referenced as a use-case for future editors and authors.

What is a "Data Citation"?

Data Citations track the provenance of a dataset giving credit to the data generator; this is in addition to any references to publications where the data are described. Data Citations are used in the text (or tables) alongside and in the same way as publication citations. In the Reference list, they include: Creators, Title, Repository, Identifier, Submission Year. More information about Data Citations is here: Here is an example of text and corresponding citations (using CP punctuation style):

"The PAGES2k Consortium (2017a) assembled a large global dataset of temperature sensitive proxy records (PAGES2k Consortium, 2017b). Among the records is the paleo-temperature reconstruction from Laguna Chepical (de Jong et al., 2016), which was described by de Jong et al. (2013)."

References

de Jong, R., von Gunten, I., Maldonado, A., and Grosjean, M.: Late Holocene summer temperatures in the central Andes reconstructed from the sediments of high-elevation Laguna Chepical, Chile (32S), Climate of the Past, 9, 1921-1932, 2013.

de Jong, R., von Gunten, I., Maldonado, A., and Grosjean, M.: Laguna Chepical summer temperature reconstruction, World Data Center for Paleoclimatology, https://www.ncdc.noaa.gov/paleo/study/20366, 2016.

PAGES 2k Consortium: A global multiproxy database for temperature reconstructions C3 CPD Interactive comment Printer-friendly version Discussion paper of the Common Era, Scientific Data, 4,170088, 2017a.

PAGES 2k Consortium: A global multiproxy database for temperature reconstructions of the Common Era, version 2.0.0, NOAA-WDS Paleoclimatology, https://www.ncdc.noaa.gov/paleo/study/21171, 2017b.
* * *

---

## Referee Comment (RC1) · Anonymous Referee #1 · 1 Jun 2018

General comments

This work proposes to study the variability of reconstructed SST (15 records), benthic d13C (26 records) and dust records (9 records) though Mid-Brunhes Transition (MBT) using a statistical characterization. It is based on the compilation of already published data and reconsideration of age-scale to use the common LR04 d18O age. No new proxy data are provided. The main finding is that the MBT was a global event but the changes were not synchronous. The authors' basic idea to use statistical characterization may have a potential. However, the same research subject with more compiled data was already treated: for instance, 46 d13C records in Lisiecki (2014) and 49 paired SST-planktonic d18O records in Shakun et al. (2015). Therefore, critical points of the present work are robustness of representability of PC1 record of each proxy and

original finding about climate mechanism inferred from the PC1. I am afraid that the authors do not succeed in these points. I will develop my concerns below.

1. Robustness of representability of compiled records The number of compiled proxy records are smaller than the previous studies. The representability of records to discuss global/regional trends is seriously questioned because of such limited data sets with heterogeneous spatial distribution. In addition, SST proxies are based on alkenone, Mg/Ca, transfer function/modern analog. These different proxies may have distinct bias because of seasonality and depth distribution in water column of proxy producers as well as proxy preservation state. Since each site is represented by one proxy, it is not clear whether the observed regional trend reflects real geographical trend or the bias related to proxy. In addition, there is no explanation about possible bias and its potential influence of extracted PC1 trend. The similar difficulty exists for dust records since this variable is estimated from dust flux, the mass accumulation rate of detrital fraction or detrital element, grain size and the concentration of detrital element. Concentration of detrital element is not always representative of dust flux since the variability of sediment density and sedimentation rate are important in certain regions. Again, possible influence of mixed indicators on dust PC1 is not discussed. The authors are careful with temporal resolution of selected records but there is no information on sedimentation rate of considered records. Bioturbation affects amplitude of variability as well as lead/lag of signals. It is not clear whether the authors applied certain criteria of sedimentation rate for their compilation. At last, the use of d18O to obtain a common age model is not sufficiently explained. It is unclear whether only benthic foraminifera d18O values were used to tune to LR04 or planktonic d18O values were also considered. Since offset between benthic and planktonic d18O may exist, the use of planktonic d18O could add further uncertainty of the representability and timing of compiled records.

Above mentioned points are examples that should be clarified to go further.

2. Original new finding of the present study in relation to climate mechanism Since
no new data are presented, the significance of this work essentially depends on new observation based on the compiled data that were not revealed by individual records and climate mechanism that can be inferred from the compilation. Unfortunately, it is difficult to identify such findings. For instance, the authors interpret d13C excursion during MIS13 is due to "a change in the carbon reservoir and not related to ocean circulation". Then, the authors propose that stronger monsoon (thus more precipitation) during MIS13 that followed by smaller ice sheets of MIS 14 contributed to more light carbon storage on continents during MIS13. It is curious that they do not refer the work by Hoogakker et al. (2006) that proposed an alternative mechanism. Hoogakker et al. (2006) treated the same theme by the compilation of surface and deep-dwelling planktonic d13C and box modelling. They suggested detailed mechanism that consists of concomitant changes in the burial fluxes of organic and inorganic carbon because of ventilation changes and/or changes in the production and export ratio. Section 4.1 should be revised considering this work. Also, the two result sections ("d13C" and "d13C gradient") should be revised because they are difficult to follow (see my specific/minor comments below). About ocean circulation changes in the Atlantic basin, there is some confusion. The authors interpret that the larger north-south latitudinal gradient of d13C during pre-MBT is as a sign of greater northward penetration of AABW thus less contribution of NADW compared to post-MBT. This interpretation is odd because the North Atlantic d13C record does not show significant change through MBT (Figure 12a). It is more reasonable to assume that the latitudinal gradient is caused by changes in water properties in the south Atlantic (Figure 12b and 12c). Indeed, reconstructed seawater Nd isotopic composition from a core in the equatorial Atlantic suggests a similar proportion of NADW during the interglacials of pre-MBT and post-MBT (Howe and Piotrowski, 2017). Therefore the authors' statement is inconsistent with that of Howe and Piotrowski (2017) that is cited in the present manuscript .

The manuscript contains 17 figures, which is too many regarding the messages. More efforts should be paid to select information to establish a coherent story.

Taken together, this manuscript would not be published with the present form. Overhaul reorganization is necessary to improve abovementioned points. Below I will cite non-exhaustive minor or specific comments if the authors consider a resubmission of the manuscript.

Minor or specific comments

Line 11. Delete "benthic oxygen isotope records" and go directly "sea level" like Chalk et al. (2017). This is because benthic d18O records contain bottom water temperature and other component not related to sea-level changes (Elderfield et al., 2012; Rohling et al., 2014).

Lines 17-18. Which physical mechanisms could create "the onset of high-amplitude variability in sea level at $\sim$ 430 ka that was preceded by changes in ice sheets during MIS 14 and 13"? This sentence is unclear.

Lines 90-95 and Figure 3. I am not convinced by the necessity to show the results of Blackman-Tukey power spectral analysis because the results of wavelet analysis are presented in Figure 5.

Lines 171-174. "Factor… spectral power". This part is unclear.

Lines 176-177. It is unclear why "d13CAtl PC2 is a record of changes in the isotopic values of the North Atlantic carbon reservoir rather than circulation changes". The result section contains interpretation that is not sufficiently explained.

Lines 191-194. In relation to the previous point, it is unclear why the residual time series (deep north Atlantic d13C – intermediate north Atlantic d13C) reflects only the relative influences of AABW and NADW in the north Atlantic. Consequently, the meaning of Figure 10 is not obvious.

Line 216. "These proxies" are unclear.

Line 264. Add reference(s) after "through a glacial cycle".

Figure 2. SST sites will be shown by distinguishing different proxies (with colour code, for example). Figure 8 would be deleted. Figure 9 is almost the same as Figure 3 of Lisiecki (2014).

Supplement: Table S1: The latitude and longitude of core CLP and Lake Baikal are missing. It is necessary to add the depth in water column at core location.

Reference list for supplement is missing.

References

Chalk, T. B., Hain, M. P., Foster, G. L., Rohling, E. J., Sexton, P. F., Badger, M. P. S., Cherry, S. G., Hasenfratz, A. P., Haug, G. H., Jaccard, S. L., Martínez-García, A., Pälike, H., Pancost, R. D., and Wilson, P. A.: Causes of ice age intensification across the Mid-Pleistocene Transition, Proceedings of the National Academy of Sciences, 114, 13114-13119, 2017. Elderfield, H., Ferretti, P., Greaves, M., Crowhurst, S., McCave, I. N., Hodell, D., and Piotrowski, A. M.: Evolution of Ocean Temperature and Ice Volume Through the Mid-Pleistocene Climate Transition, Science, 337, 704-709, 2012. Hoogakker, B. A. A., Rohling, E. J., Palmer, M. R., Tyrrell, T., and Rothwell, R. G.: Underlying causes for long-term global ocean $\delta$13C fluctuations over the last 1.20 Myr, Earth and Planetary Science Letters, 248, 15-29, 2006. Howe, J. N. W. and Piotrowski, A. M.: Atlantic deep water provenance decoupled from atmospheric CO2 concentration during the lukewarm interglacials, Nature Communications, 8, 2003, 2017. Lisiecki, L. E.: Atlantic overturning responses to obliquity and precession over the last 3 Myr, Paleoceanography, 29, 71-86, 2014. Rohling, E. J., Foster, G. L., Grant, K. M., Marino, G., Roberts, A. P., Tamisiea, M. E., and Williams, F.: Sea-level and deep-sea-temperature variability over the past 5.3 million years, Nature, 508, 477-482, 2014. Shakun, J. D., Lea, D. W., Lisiecki, L. E., and Raymo, M. E.: An 800-kyr record of global surface ocean and implications for ice volume-temperature coupling, Earth and Planetary Science Letters, 426, 58-68, 2015.

---

## Referee Comment (RC2) · Anonymous Referee #2 · 23 Jul 2018

Aaron M. Barth et al, present an interesting study and well written manuscript, but the large number of figures (17 figures) make the reading difficult. This work addresses a statistical characterization of changes occurring during Mid-Brunhes Transition (MBT) over the last 800kyr. This work is based on already published data of SST, benthic d13C and dust records from Atlantic, Pacific and Indian Ocean. The main proposal of this work is to demonstrated that MBT is a global event. However the representability of the selected recorded in order to discuss global /regional patterns is not clear. There are several record, some of them on North Atlantic that they show different pattern at least on SST trend and warmer interglacial are record on pre-MBT interval as also was mentioned on previous review papers (Pages, 2016). So should be very interesting to analyze the differences observed on these patterns and clear define what the global

concept is, if it is the patterns similar with the Ice core records or the regional forcing are always imprinted on our climate records. It would be good to add in the introduction a few sentences to explain why it was selected these sites. Special focus was been done on MIS 14 and 13 and the interconnections between strong Asian monsoon on MIS 13 and the followed weak glaciation MIS14, however this assumption does not take in account the ventilation and the changes in the d13C on North Atlantic during this interval as also involved on the main changes at the CO2 and SST pattern. As the major comments on the manuscript I consider that would be good to add other sites and integrated different patterns on this global concept.

Pages, P.I.W.G.o., 2016. Interglacials of the last 800,000 years. Reviews of Geophysics 54, 2015RG000482.

---

## Editor Comment (EC1) · R. Barnett (Editor) · 1 Aug 2018

Dear Dr Barth,

Thank you for your submission of 'Climate evolution across the Mid-Brunhes Transition' to the special issue 'Global Challenges for our Common Future: a paleoscience perspective' for the journal Climate of the Past. The final comments from the reviewers have now been published in the interactive discussion section of your submission. In considering the comments from the reviewers, we encourage you to submit a revised version of your manuscript following major revision. We draw your particular attention to the two main points raised by Reviewer #1. Please can you provide clarification on the novelty of the study and for the choice of datasets analysed. Both reviewers also

demonstrate awareness for the number of figures presented in the manuscript. Please consider reducing the number of figures required to support the primary argument of the manuscript. Following revisions, we feel that this submission will provide valuable and contemporary analysis and commentary on the synchronicity of the Mid-Brunhes Transition and will be thoroughly suitable for publication in this journal.

If you choose to submit a revised manuscript, please can you respond to the reviewer comments in turn and upload these as author comments within the submission portal. Please could you also identify changes made within the manuscript when uploading the revised version of your manuscript.

To accord with the PAGES initiative on data stewardship, please could you meet the requests made by the PAGES Data Review Team as found in the interactive discussion. This will be necessary prior to the publication of your manuscript.

Thank you again for your submission to this journal.

Kind regards,

Rob Barnett

on behalf of the editorial team, Mike Evans, Marie-France Loutre, Rob Barnett
* * *

---

## Author Comment (AC1) · 27 Sep 2018

Response to PAGES Data Review Team – Climate of the Past

For this paper:

All papers submitted to "Climate of the Past" must include a Data Availability section that details the location of the data that were used as input to the study, including previously published data that were used for comparison purposes, and the data that were generated by the study.

Input Data (used for analyses) – Data citations are needed for all time-series data that were used in this study. Add a column to Table S1 to list the persistent identifier (doi or URL from NOAA-Paleo) for each record. We recommend that Table S1 be moved to the main text so that data generators are acknowledged more prominently. Change the "researcher" column in Table S1 to "reference" and include the bibliographic citation in the References. If the data are not already available in long-standing data repositories, then we ask the authors to work with the data generators (if possible) to rescue the data and transfer them along with essential metadata to a data repository (NOAA-Paleo, PANGEA, or other registered at re3data.org). Once curated in a pubic repository, each dataset will be assigned a persistent identifier, which is then cited in this paper to credit the original data generators, in addition to the bibliographic reference.

*The suggested changes were made to the table and it has been moved to the main text out of the supplement. All relevant doi/url were added to each record where possible. Some data sets were received directly from the author and not available in an online data repository. Attempts are being made to make the data available to all.*

In this study, the input data were modified for some records to adjust the age scale, and all time series were interpolated at 2-kyr intervals. To enable others to reproduce the primary results of this study, and to avoid losing this excellent compilation of proxy data, please transfer this entire modified dataset to a public repository, along with basic metadata for each record. The compilation should include the data and bibliographic citations for each of the constituent records, and the package (with its own persistent identifier) should be attributed to the authors of this study as an outcome of this synthesis.

*The input data for our analysis were compiled into a single file with metadata descriptions and proper references, and uploaded to Pangaea. The data set is currently being checked and processed.*

Input Data (used for comparison) - Please cite and reference the source of the insolation curves shown in Figs. 1a,b,c, 14a. Because these data are used to compare rather than generate the primary outcome comes of the study, we suggest that a bibliographic reference to the data would suffice, although the authors are encouraged to include a data citation in addition. If, however, the time series were generated by the authors using online software, then both the data behind the calculator and the developer of the software should be cited.

*The insolation curves were calculated using the Analyseries program and is now referenced in the bibliography as:*

*Paillard, D., Labeyrie, L., and Yiou, P.: Macintosh program performs time-series analysis, Eos Trans. AGU, 77, 1996.*

*Proper citations have also been added to the figure captions.*

Output Data (analytical) – Please prepare and upload to a registered data repository all of the data and metadata resulting from the primary analyses in the manuscript. This includes the time series of record stacks (Fig. 9), and derived temperature gradients (Fig. 12). Once assigned a persistent identifier (doi or URL from NOAA-Paleo), include the data citation for these files in the manuscript. Most data repositories use landing pages that collect the individual data files under a single identifier.

*The input data for our analysis were compiled into a single file with metadata descriptions and proper references, and uploaded to Pangaea. The data set is currently being checked and processed.*

Output Data (statistical) – For the results of the statistical analyses including PCs (Fig. 4, 6b, 7, 8, 13a) spectral analyses (Figs. 3, 5) and d13C contour plots (Figs. 11, 15), please ensure that the statistical packages that were used to produce the plots are cited and fully referenced in the manuscript with sufficient details so that the plots could be replicated in future. We encourage the authors to upload the results of the statistical analyses to a repository (as above), but leave it to them to determine the likely future utility of the digital result of the statistical analyses relative to the effort involved to curate them.

*We note that the ARAND software package for our statistical analyses has been cited as:*

*"Howell, P., Pisias, N. G., Ballance, J., Baughman, J., and Ochs, L.: ARAND Time-Series Analysis Software, Brown University, Providence, RI, 2006."*

*Ocean Data View that was used to create the contour plots is now cited as:*

*"Schlitzer, Reiner: AWI's Oecan Data View (ODV), Alfred Wegener Institute for Polar and Marine Research, 4.5.0."*

---

## Author Comment (AC2) · 27 Sep 2018

We would like to thank the reviewer for the constructive comments that helped improve the clarity and value of our paper. Please see our comments attached as a supplement including a revised version of the manuscript. figures, and supplemental materials.

Please also note the supplement to this comment: https://www.clim-past-discuss.net/cp-2018-20/cp-2018-20-AC2-supplement.zip

---

## Author Response (AR1)

Comments to the Author:

Dear Dr Barth,

Thank you for submitting the revised manuscript for cp-2018-20 Climate Evolution Across the Mid-Brunhes Transition and for responding to reviewer comments and requests from the Data Review Team. We will be happy to accept this manuscript for publication in Climate of the Past following minor revisions. Please can you address the following comments and resubmit your manuscript with changes annotated:

*Thank you. We were all very pleased to hear this good news, and happy to resubmit the suggested changes outlined below. Please let us know what else we can do moving forward.*

i) Please include a data availability section at the end of the manuscript in line with the data policy of the journal: https://www.climate-of-the-past.net/about/data_policy.html

*A Data Inventory section is now included in the manuscript.*

ii) Please include all analytical output data from this study within the PANGAEA repository related to this submission (https://doi.pangaea.de/10.1594/PANGAEA.894896) and include this DOI in the data availability section

*Analytical output data is now uploaded to PANGAEA and is awaiting final acceptance as part of the associated repository (https://doi.pangaea.de/10.1594/PANGAEA.894896).*

iii) Please, where possible, complete the list of persistent identifiers for input data associated with this study. Where none are available, please outline efforts made to make these data available

*The few remaining data sets without a persistent identifier have been acknowledged in the manuscript. A note is added indicating our willingness to share the data for those interested.*

iv) please include full methodological details in statistical procedures applied in the study so that the study can be fully replicated. In particular, please provide greater detail in sections 2.5, 2.6 and 2.7 and include citation where appropriate.

*We provide more clarity in our exact approach to the analyses and added citations where appropriate.*

v) please provide citations and references for data in figures 1a,b,c and 11a

*Citations are now included.*

Thank you for your continued efforts in seeing this submission through to publication. I apologise for delays in this process. I assure you that I will respond to queries and manuscript submission as soon as they happen.

All the best,

Rob Barnett

[revised manuscript text omitted]